# Direct Consistency Optimization for Robust Customization of Text-to-Image Diffusion Models

**Kyungmin Lee**[1]    **Sangkyung Kwak**[1]    **Kihyuk Sohn**[2]*    **Jinwoo Shin**[1]
[1]KAIST    [2]Meta Reality Labs

{kyungmnlee, skkwak9806, jinwoos}@kaist.ac.kr
kihyuk.sohn@gmail.com

## Abstract

Text-to-image (T2I) diffusion models, when fine-tuned on a few personal images, can generate visuals with a high degree of consistency. However, such fine-tuned models are not robust; they often fail to compose with concepts of pretrained model or other fine-tuned models. To address this, we propose a novel fine-tuning objective, dubbed *Direct Consistency Optimization*, which controls the deviation between fine-tuning and pretrained models to retain the pretrained knowledge during fine-tuning. Through extensive experiments on subject and style customization, we demonstrate that our method positions itself on a superior Pareto frontier between subject (or style) consistency and image-text alignment over all previous baselines; it not only outperforms regular fine-tuning objective in image-text alignment, but also shows higher fidelity to the reference images than the method that fine-tunes with additional prior dataset. More importantly, the models fine-tuned with our method can be merged without interference, allowing us to generate custom subjects in a custom style by composing separately customized subject and style models. Notably, we show that our approach achieves better prompt fidelity and subject fidelity than those post-optimized for merging regular fine-tuned models.[2]

## 1   Introduction

Text-to-image (T2I) models are for image generation guided by natural language prompts and have seen rapid progress in recent years [1–8]. The compositional nature of the natural language has enabled the creation of novel images, which compose multiple subjects with varying attributes in different backgrounds or styles. However, the ambiguity of natural language in describing the visual world makes it difficult to create an image of a specific subject, style, interaction, or background.

To overcome the lack of accuracy in natural language, there has been an emerging interest in teaching the pretrained T2I models new concepts, such as subject [9, 10], style [11], interaction [12], or background [13], whose precise visual description is given by a small set of reference images. As proposed in DreamBooth [10], the fundamental idea is to fine-tune the pretrained T2I model on a few images describing a new concept. The adoption of LoRA [14] or adapter fine-tuning [15] to T2I models has made the process more accessible, fast and economical [16, 11]. Once fine-tuned, the model can generate images by composing a new concept (*e.g.*, my subject) and the knowledge of the pretrained model (*e.g.*, background, style). While these methods have shown great success, they still suffer from reduced textual alignment and compositional generation capability [17], which is problematic particularly when the number of reference images is small, *e.g.*, one or two.

---

*Work done while at Google Research.

[2]**Project page**: https://dco-t2i.github.io

38th Conference on Neural Information Processing Systems (NeurIPS 2024).

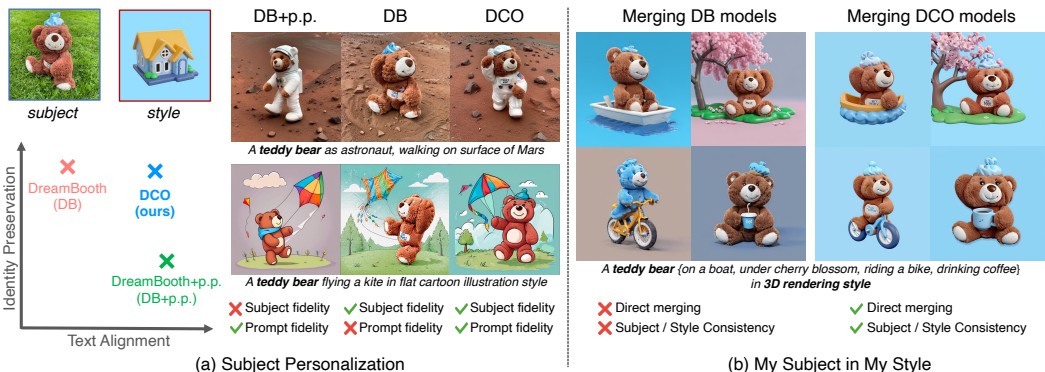

Figure 1: **Overview.** (a) *Direct Consistency Optimization* (DCO) pushes the Pareto frontier between prompt fidelity and subject fidelity towards upper-right over DreamBooth [10], and with prior preservation loss (DreamBooth+p.p.). DCO improves generating custom subject with various visual attributes (*e.g.*, astronaut outfits and background of Mars), or various styles that pretrained model knows (*e.g.*, flat cartoon illustration style). (b) The customized subject and style models fine-tuned with DCO can be merged as is, allowing us to generate *my subject in my style* [11].

We hypothesize that the limitation comes from the knowledge forgetting during low-shot fine-tuning, *i.e.*, it fails to recontextualize new concepts with known concepts of pretrained models. Since fine-tuning with regular diffusion training objective suffer from overfitting, DreamBooth [10] attempted to tackle this issue by adding class-specific prior dataset for training dataset to preserve the prior knowledge (*i.e.*, prior preservation loss). However, while this helps retaining the generalization capability, it comes at the cost of less subject fidelity (*e.g.*, see Fig. 3). Especially, the loss becomes more significant, when the number of reference images become smaller [18].

**Contribution.** In this paper, we propose a more principled method to mitigate the forgetting behavior of low-shot fine-tuning of T2I diffusion models without using any additional data. Our method, coined *Direct Consistency Optimization* (DCO), fine-tunes T2I diffusion models by controlling the deviation between fine-tuning and pretrained model to be as minimal as possible in learning new concept. In specific, we show that our objective is equivalent to learning an implicit reward function that is a solution of constrained policy optimization problem [19, 20], which amounts for the consistency between generated images and reference images. Furthermore, we propose consistency guidance sampling which utilizes learned consistency function during inference, so that one can control the tradeoff between subject consistency and textual alignment (*e.g.*, see Fig. 5).

We conduct an extensive empirical study on the personalization of T2I diffusion models (see Fig. 1). We show that the proposed method improves upon baselines that uses regular fine-tuning objective or using prior preservation loss [10] in generation of custom subject with various visual attributes or with the known style of pretrained model. To be specific, we show that our method positions on the superior Pareto frontier than baselines (*e.g.*, see Fig. 5a). Moreover, our approach is applied to style personalization [11] and the composition of separately fine-tuned subject and style T2I models to generate an image of *my subject and my style* [11]. Notably, direct merging of DCO fine-tuned models outperforms merging models fine-tuned with regular diffusion loss even using post-optimization method, *e.g.*, ZipLoRA [21], in terms of prompt fidelity and subject fidelity. We believe that the proposed method would enjoy a broader usage in customization diffusion models in various domains.

## 2 Related Work

We briefly review the relevant literatures on personalized T2I synthesis and fine-tuning of T2I diffusion models, especially using rewards. We provide more comprehensive review and discussion on related works in Appendix C.

**Personalized T2I synthesis.** Several works have shown the promise of personalized T2I synthesis from a few images [9, 10, 22, 23, 11]. To improve efficiency, parameter efficient fine-tuning (PEFT) methods such as soft prompt tuning [9, 24–27], LoRA [14, 28] and adapter tuning [15, 11], have been proposed. To preserve the prior knowledge, Ruiz et al. [10] proposed prior preservation loss, which additionally fine-tunes on a class-specific prior dataset synthesized from pretrained T2I model. We

present a different approach to preserve prior knowledge by regularizing the model with pretrained model without using any prior dataset.

**Fine-tuning T2I diffusion models with rewards.** A line of works has studied fine-tuning T2I diffusion models using reward models such as human preferences [29–31] or aesthetic scores [32]. Those methods use auxiliary reward models to update diffusion models by weighting with rewards [29], use reinforcement learning algorithms [33, 34], or differentiate through the reward models [35, 36]. On the other hand, Wallace et al. [37] have adapted direct preference optimizaton [38] to fine-tune T2I models with paired preference dataset which do not require explicit reward models. To the best of our knowledge, this paper is the first to study consistency as a reward function for T2I personalization.

## 3 Preliminaries

**Diffusion models.** For a data point $\mathbf{x}$, let $q(\mathbf{x})$ be the density of data distribution and $p_\theta(\mathbf{x})$ be a generative model parameterized by $\theta$ that approximates $q$. Given $\mathbf{x} \sim q(\mathbf{x})$, the diffusion model considers a series of latent variables $\mathbf{z}_t$ for timesteps $t \in [0, 1]$. The forward process of diffusion model forms a conditional distribution $q(\mathbf{z}_t|\mathbf{x})$, and the marginal distribution is given by $\mathbf{z}_t = \alpha_t \mathbf{x} + \sigma_t \boldsymbol{\epsilon}$, where $\boldsymbol{\epsilon} \sim \mathcal{N}(\mathbf{0}, \mathbf{I})$, and $\alpha_t, \sigma_t$ are noise scheduling functions. We denote $\lambda_t = \log(\alpha_t^2/\sigma_t^2)$ log signal-to-noise ratio (log-SNR), which is a decreasing function of $t$. For large enough $\lambda_1$, $\mathbf{z}_1$ is indistinguishable from pure Gaussian noise (*i.e.*, $p(\mathbf{z}_1) \approx \mathcal{N}(\mathbf{0}, \mathbf{I})$), and for small enough $\lambda_0$, the $\mathbf{z}_0$ is identical to the data $\mathbf{x}$. The generative process of diffusion models starts off from a random noise $\mathbf{x}_1 \sim \mathcal{N}(\mathbf{0}, \mathbf{I})$, and sequentially denoise it to recover $\mathbf{z}_0$. In theory, the generative sampling process is governed by solving SDE [39, 40] or probability flow ODE [41, 42], using the score networks $\mathbf{s}_\theta(\mathbf{z}_t; t)$ that approximates the score function of marginal distribution $\nabla \log q(\mathbf{z}_t)$. When training diffusion models, it is common practice to parameterize score network using the noise-prediction model [40], *i.e.*, $\mathbf{s}_\theta(\mathbf{z}; t) = -\boldsymbol{\epsilon}_\theta(\mathbf{z}; t)/\sigma_t$. Then the training objective of diffusion model is given by

$$\mathcal{L}_{\boldsymbol{\epsilon}}(\theta; \mathbf{x}) = \mathbb{E}_{t \sim \mathcal{U}(0,1), \boldsymbol{\epsilon} \sim \mathcal{N}(\mathbf{0}, \mathbf{I})}\left[ -\tfrac{1}{2} w_t \lambda_t' \|\boldsymbol{\epsilon}_\theta(\mathbf{z}_t; t) - \boldsymbol{\epsilon}\|_2^2 \right], \quad (1)$$

where $w_t$ is a weighting function and $\lambda_t'$ is a time-derivative of $\lambda_t$. Note that Eq. (1) is a generalized weighted noise-prediction loss, which includes $\boldsymbol{\epsilon}$-prediction loss [40] when $-\tfrac{1}{2} w_t \lambda_t' = 1$.

**Text-to-Image diffusion models.** Text-to-Image (T2I) diffusion models [43, 2, 4] are diffusion models of an image conditioned on text, which is often processed into embeddings using the pretrained text encoders, such as T5 [44] or CLIP [45]. Given a dataset $(\mathbf{x}, \mathbf{c}) \sim \mathcal{D}$ of paired image $\mathbf{x}$ and text prompt $\mathbf{c}$, the training loss $\mathcal{L}_{\mathrm{DM}}(\theta; \mathcal{D})$ for T2I diffusion models is given by $\mathcal{L}_{\mathrm{DM}}(\theta; \mathcal{D}) = \mathbb{E}_{(\mathbf{x}, \mathbf{c}) \sim \mathcal{D}}[\mathcal{L}_{\boldsymbol{\epsilon}}(\theta; \mathbf{x}, \mathbf{c})]$, with text conditional noise-prediction model $\boldsymbol{\epsilon}_\theta(\mathbf{z}; \mathbf{c}, t)$. T2I diffusion models are often trained with classifier-free guidance (CFG) [46], which jointly learns unconditional and conditional models, and interpolates them during inference. The predicted noise with CFG scale $\omega \geqslant 1$ is given as:

$$\hat{\boldsymbol{\epsilon}}_\theta(\mathbf{z}_t; \mathbf{c}, t) = \omega\big(\boldsymbol{\epsilon}_\theta(\mathbf{z}_t; \mathbf{c}, t) - \boldsymbol{\epsilon}_\theta(\mathbf{z}_t; t)\big) + \boldsymbol{\epsilon}_\theta(\mathbf{z}_t; t), \quad (2)$$

where $\boldsymbol{\epsilon}_\theta(\mathbf{z}_t; t)$ denotes the unconditional noise prediction, *e.g.*, with null text conditioning. It is known that higher CFG scale $\omega$ improves the image-text alignment at the cost of the image fidelity.

**Personalizing T2I models.** Recent works have shown the potential for personalization of T2I models by fine-tuning the T2I diffusion models on a few samples. DreamBooth [10] optimizes the diffusion model on a few subject images accompanied with a compact caption composed of rare token identifier and class noun. While fine-tuning with regular diffusion loss (*i.e.*, $\mathcal{L}_{\mathrm{DM}}$) works well, the authors proposed the so-called prior preservation loss to retain the prior knowledge of the pretrained model. This is achieved by optimizing the model with auxiliary training images of the same class to the subject of interest. Formally, given reference dataset $\mathcal{D}_{\mathrm{ref}}$ and prior dataset $\mathcal{D}_{\mathrm{prior}}$, the training loss of DreamBooth with prior preservation loss (DB+p.p.) with hyperparameter $\lambda_{\mathrm{prior}} > 0$ is given by

$$\mathcal{L}_{\mathrm{DB+p.p.}}(\theta) = \mathcal{L}_{\mathrm{DM}}(\theta; \mathcal{D}_{\mathrm{ref}}) + \lambda_{\mathrm{prior}} \mathcal{L}_{\mathrm{DM}}(\theta; \mathcal{D}_{\mathrm{prior}}). \quad (3)$$

**Parameter efficient fine-tuning (PEFT).** In practice, parameter-efficient fine-tuning (PEFT) methods are combined with DreamBooth to enable fast and memory-efficient adaptation of diffusion models. In particular, low-rank adaptation (LoRA) [14] is a popular choice, where it fine-tunes the residuals $\Delta W \in \mathbb{R}^{n \times m}$ of weight matrix $W \in \mathbb{R}^{n \times m}$ with low-rank decomposition $\Delta W = AB$ for $A \in \mathbb{R}^{n \times r}$ and $B \in \mathbb{R}^{r \times m}$ with rank $r \ll \min\{n, m\}$. Alternatively, Textual Inversion (TI) [9] introduces a new token and corresponding textual embedding $\mathbf{v}$ to represent the concept. Then, TI optimizes textual embedding by solving $\mathbf{v}^* = \arg\min_{\mathbf{v}} \mathcal{L}_{\mathrm{DM}}(\mathbf{v}; \mathcal{D}_{\mathrm{ref}})$.

| **Algorithm 1** Regular fine-tuning | **Algorithm 2** Fine-tuning with DCO loss |
|---|---|
| **Require:** Dataset $\mathcal{D}_{\text{ref}}$, fine-tuning model $\boldsymbol{\epsilon}_\theta$, learning rate $\eta > 0$ | **Require:** Dataset $\mathcal{D}_{\text{ref}}$, fine-tuning model $\boldsymbol{\epsilon}_\theta$, pretrained model $\boldsymbol{\epsilon}_\phi$, temperature $\beta_t > 0$, learning rate $\eta > 0$ |
| 1: **while** not converged **do** | 1: **while** not converged **do** |
| 2:    Sample $(\mathbf{x}, \mathbf{c}) \sim \mathcal{D}_{\text{ref}}$ | 2:    Sample $(\mathbf{x}, \mathbf{c}) \sim \mathcal{D}_{\text{ref}}$ |
| 3:    Sample $\boldsymbol{\epsilon} \sim \mathcal{N}(\mathbf{0}, \mathbf{I})$ | 3:    Sample $\boldsymbol{\epsilon} \sim \mathcal{N}(\mathbf{0}, \mathbf{I})$ |
| 4:    Sample $t \sim \mathcal{U}(0, 1)$ | 4:    Sample $t \sim \mathcal{U}(0, 1)$ |
| 5:    $\mathbf{z}_t \leftarrow \alpha_t \mathbf{x} + \sigma_t \boldsymbol{\epsilon}$ | 5:    $\mathbf{z}_t \leftarrow \alpha_t \mathbf{x} + \sigma_t \boldsymbol{\epsilon}$ |
| 6:    $\mathcal{L}_{\text{DM}}(\theta) \leftarrow \|\boldsymbol{\epsilon}_\theta(\mathbf{z}_t; c, t) - \boldsymbol{\epsilon}\|_2^2$ | 6:    $\ell(\theta) \leftarrow \|\boldsymbol{\epsilon}_\theta(\mathbf{z}_t; c, t) - \boldsymbol{\epsilon}\|_2^2$ |
| 7:    Update $\theta \leftarrow \theta - \eta \nabla_\theta \mathcal{L}_{\text{DM}}(\theta)$ | 7:    $\ell(\phi) \leftarrow \|\boldsymbol{\epsilon}_\phi(\mathbf{z}_t; c, t) - \boldsymbol{\epsilon}\|_2^2$ (no gradient) |
| 8: **end while** | 8:    $\mathcal{L}_{\text{DCO}}(\theta) \leftarrow -\log \sigma\big(-\beta_t\big(\ell(\theta) - \ell(\phi)\big)\big)$ |
|  | 9:    Update $\theta \leftarrow \theta - \eta \nabla_\theta \mathcal{L}_{\text{DCO}}(\theta)$ |
|  | 10: **end while** |

## 4 Method

In this section, we introduce our method for T2I personalization. For presentation clarity, we focus on our demonstration with the subject customization [10], but the method can be applied to a broader context of personalization, such as style [11]. Throughout the paper, let us denote $\boldsymbol{\epsilon}_\theta, p_\theta$ and $\boldsymbol{\epsilon}_\phi, p_\phi$ the noise-prediction model and density for each fine-tuning and pretrained diffusion model, respectively.

### 4.1 Direct Consistency Optimization

**Problem setup.** While fine-tuning based T2I personalization methods have shown great success [10, 23, 16], it is shown that the generation quality heavily depends on the model's fitness. For example, the model suffers from image-text alignment when the model overfits to few images used for fine-tuning, making it difficult to generate images with varying attributes around the subject. On the other hand, the model cannot generate consistent subject images when the model underfits. To find the right balance between overfit vs. underfit, certain heuristics such as early stopping or training with an additional prior dataset (*i.e.*, prior-preservation loss) have been popularly used.

**Our approach.** We devise an efficient training objective that seeks for minimal improvement on the ELBO of fine-tuning model $p_\theta$ over the reference model $p_\phi$. Specifically, we seek for $\theta$ satisfying $D_{\text{KL}}\big(q(\mathbf{x})\|p_\theta(\mathbf{x})\big) < D_{\text{KL}}\big(q(\mathbf{x})\|p_\phi(\mathbf{x})\big)$, while two quantities are still not far from each other. Since computing the likelihood is intractable for diffusion models, we consider the variational bound over sequence of latents $\mathbf{z}_{0:1}$ following [47, 48]. To this end, let us define the deviation as follows

$$\Delta(p_\theta, p_\phi; \mathbf{x}, \mathbf{c}) := D_{\text{KL}}\big(q(\mathbf{z}_{0:1}|\mathbf{x}) \,\|\, p_\phi(\mathbf{z}_{0:1}|\mathbf{c})\big) - D_{\text{KL}}\big(q(\mathbf{z}_{0:1}|\mathbf{x}) \,\|\, p_\theta(\mathbf{z}_{0:1}|\mathbf{c})\big), \qquad (4)$$

where the KL divergence takes expectation over $\boldsymbol{\epsilon} \sim \mathcal{N}(\mathbf{0}, \mathbf{I})$ that generates $\mathbf{z}_t = \alpha_t \mathbf{x} + \sigma_t \boldsymbol{\epsilon}$ for $t \in (0, 1)$. To enforce the positiveness of $\Delta(p_\theta, p_\phi)$, we propose following log-sigmoid loss function:

$$\mathcal{L}_\Delta(\theta; \mathbf{x}, \mathbf{c}) := -\log \sigma\big(\beta \, \Delta(p_\theta, p_\phi; \mathbf{x}, \mathbf{c})\big), \qquad (5)$$

where $\sigma(u) = (1 + \exp(-u))^{-1}$ and $\beta > 0$ is a temperature that controls the deviation.

**Direct Consistency Optimization.** Now we show that Eq. (5) is equivalent to optimizing reward function derived from the solution of constrained policy optimization problem [19, 20]. Suppose $f(\mathbf{x}, \mathbf{c}; \mathcal{D}_{\text{ref}}) := f(\mathbf{x}, \mathbf{c})$ is a function that measures the consistency between image $\mathbf{x}$ and reference dataset $\mathcal{D}_{\text{ref}}$ given the prompt $\mathbf{c}$. We opt to find $\theta$ that maximizes the consistency of generated sample $\mathbf{x} \sim p_\theta(\mathbf{x}|\mathbf{c})$, while penalizing the deviation from the pretrained model $p_\phi$. This can be written by

$$\max_\theta \ \mathbb{E}_{\mathbf{c}, \mathbf{x} \sim p_\theta(\mathbf{x}|\mathbf{c})}\big[f(\mathbf{x}, \mathbf{c})\big] - \beta D_{\text{KL}}(p_\theta(\cdot|\mathbf{c}) \,\|\, p_\phi(\cdot|\mathbf{c})), \qquad (6)$$

where $\beta > 0$ is a temperature that controls the deviation from pretrained model. Since the likelihood of diffusion model is intractable, we consider a consistency function on the latent variables $\mathbf{z}_{0:1}$, and solve following relaxation of Eq. (6):

$$\max_\theta \ \mathbb{E}_{\mathbf{c}, \mathbf{z}_{0:1} \sim p_\theta(\mathbf{z}_{0:1}|\mathbf{c})}\big[f(\mathbf{z}_{0:1}, \mathbf{c})\big] - \beta D_{\text{KL}}\big(p_\theta(\mathbf{z}_{0:1}|\mathbf{c}) \,\|\, p_\phi(\mathbf{z}_{0:1}|\mathbf{c})\big). \qquad (7)$$

Note that the optimal solution to Eq. (7) satisfies

$$p_\theta(\mathbf{z}_{0:1}|\mathbf{c}) \propto p_\phi(\mathbf{z}_{0:1}|\mathbf{c}) \exp\left(f(\mathbf{z}_{0:1}, \mathbf{c})/\beta\right). \tag{8}$$

Then, we show that the deviation $\Delta(p_\theta, p_\phi)$ in Eq. (4) is equivalent to the expected consistency over $q(\mathbf{z}_{0:1}|\mathbf{x})$ up to constant, *i.e.*, the following holds (see Appendix A for derivation):

$$\mathbb{E}_{q(\mathbf{z}_{0:1}|\mathbf{x})}[f(\mathbf{z}_{0:1}, \mathbf{c})] = \beta\Delta(p_\theta, p_\phi; \mathbf{x}, \mathbf{c}) + C, \tag{9}$$

for some constant $C$ that does not depend on $\mathbf{x}$. Eq. (9) shows that our training objective can be regarded as optimizing the consistency function with respect to the reference dataset $\mathcal{D}_{\text{ref}}$. Remark that one can define explicit consistency function (reward) and fine-tune the diffusion model with such function, *e.g.*, by using reward-weighted regression (RWR) [49, 29] or using reinforcement learning (RL) [33, 34]. However, those consistency function for personalized images is hard to obtain as we only have few samples, and RL fine-tuning of diffusion models are expensive. We bypass these issues by directly fine-tuning models with an implicit reward function, similar to those by Rafailov et al. [38] and Wallace et al. [37]. Thus, we name our approach *Direct Consistency Optimization* (DCO), as we directly optimize the consistency function in fine-tuning diffusion models.

## 4.2 Implementation and Analysis

**Implementation.** Note that computing the term for Eq. (4) is expensive as it requires computation of likelihood over all $t \in [0, 1]$. To this end, we derive an upper bound of Eq. (5) that allows efficient implementation using $\epsilon$-prediction loss. In Appendix A, we show that the deviation Eq. (4) can be expressed by the difference between noise-prediction errors of fine-tuning and pretrained model:

$$\Delta(p_\theta, p_\phi) = \tfrac{1}{2}\mathbb{E}_{t\sim\mathcal{U}(0,1),\epsilon\sim\mathcal{N}(\mathbf{0},\mathbf{I})}\left[\lambda'_t\left(\|\epsilon_\theta(\mathbf{z}_t; \mathbf{c}, t) - \epsilon\|_2^2 - \|\epsilon_\phi(\mathbf{z}_t; \mathbf{c}, t) - \epsilon\|_2^2\right)\right]. \tag{10}$$

Then by plugging Eq. (10) into Eq. (5) and taking out the expectation outside of logarithm function using Jensen's inequality, we have our final DCO loss at $(\mathbf{x}, \mathbf{c}) \sim \mathcal{D}_{\text{ref}}$ defined as follows:

$$\mathcal{L}_{\text{DCO}}(\theta; \mathbf{x}, \mathbf{c}) = \mathbb{E}_{t,\epsilon}\left[-\log\sigma\left(-\beta_t(\|\epsilon_\theta(\mathbf{z}_t; \mathbf{c}, t) - \epsilon\|_2^2 - \|\epsilon_\phi(\mathbf{z}_t; \mathbf{c}, t) - \epsilon\|_2^2)\right)\right], \tag{11}$$

where $t \sim \mathcal{U}(0, 1)$, $\epsilon \sim \mathcal{N}(\mathbf{0}, \mathbf{I})$, and $\beta_t = -\tfrac{1}{2}\beta\lambda'_t$ [3]. We use $\mathcal{L}_{\text{DCO}}$ in our experiments, which is as practical and easy to implement as regular training objective (*i.e.*, $\mathcal{L}_{\text{DM}}$). Algorithm 2 and Algorithm 1 present DCO fine-tuning and regular fine-tuning side-by-side, with differences colored in red.

**Gradient analysis of DCO loss.** We provide a gradient analysis of DCO loss in fine-tuning diffusion models to better understand its effect. Given a data pair $(\mathbf{x}, \mathbf{c}) \sim \mathcal{D}_{\text{ref}}$, $\epsilon \sim \mathcal{N}(\mathbf{0}, \mathbf{I})$ and $t \in \mathcal{U}(0, 1)$, the gradient of DCO loss with respect to parameter $\theta$ is given as follows:

$$\nabla_\theta\mathcal{L}_{\text{DCO}}(\theta) \propto (1 - \sigma(d_t))\nabla_\theta\|\epsilon_\theta(\mathbf{z}_t; \mathbf{c}, t) - \epsilon\|_2^2, \tag{12}$$

where $d_t = -\beta_t(\|\epsilon_\theta(\mathbf{z}_t; \mathbf{c}, t) - \epsilon\|_2^2 - \|\epsilon_\phi(\mathbf{z}_t; \mathbf{c}, t) - \epsilon\|_2^2)$ with stop-gradient. Remark that Eq. (12) is identical to the gradient of diffusion loss (*i.e.*, $\mathcal{L}_{\text{DM}}$), except that is scaled by $1 - \sigma(d_t)$, which measures the incorrect reward modeling. In other words, DCO loss implicitly performs an adaptive loss weighting by computing deviation from the pretrained model; if the deviation between fine-tuning and pretrained models are large, it abstains update.

**Comparison to prior preservation loss.** While the prior preservation loss [10] in Eq. (3) has a similar motivation to DCO loss, they work in very different ways. To elaborate, DCO directly regularizes the KL divergence with respect to the samples in $\mathcal{D}_{\text{ref}}$, while prior preservation loss does not impose regularization for the reference data. While prior preservation loss may enhance the composition ability, fine-tuning on auxiliary samples from $\mathcal{D}_{\text{prior}}$ often causes undesirable model shift, losing consistency to the pretrained model (*e.g.*, Fig. 3). On the other hand, DCO is free from such an issue, as it does not require any auxiliary samples besides the reference dataset.

## 4.3 Consistency Guidance Sampling

During inference, it is common practice to use classifier-free guidance to control text conditioning. Similarly, to gain control over the consistency, we propose *consistency guidance sampling*, which is

---

[3]Since $\lambda_t$ is a decreasing function of $t$, $\lambda'_t < 0$. Thus, we use $\beta_t = -\tfrac{1}{2}\beta\lambda'_t$ to enusre $\beta_t > 0$.

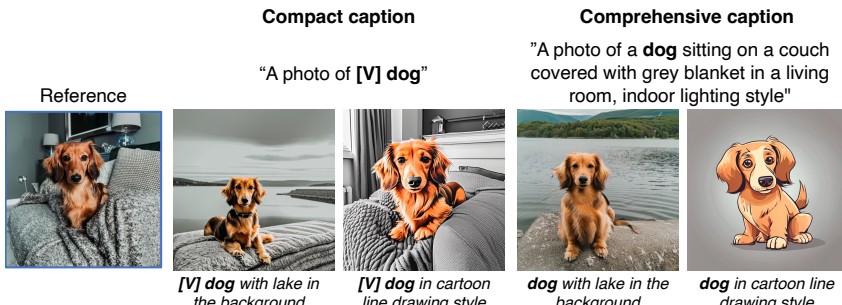

Figure 2: **Comprehensive caption.** We provide examples of compact caption [10] and our comprehensive caption (top row) and generated images from each method (bottom row). The model fine-tuned with compact caption (left) generates images of a dog sitting on a couch though asked to be on the lake. Our comprehensive caption (right) effectively disentangles unwanted attributes, generating images that follow text prompts more faithfully.

an additional guidance from implicitly learned consistency function. Specifically, we decompose the guidance term with respect to the text and consistency as follows:

$$\hat{\boldsymbol{\epsilon}}(\mathbf{z}_t; \mathbf{c}, t) = \boldsymbol{\epsilon}_\phi(\mathbf{z}_t, t) + \omega_{\text{text}} \left( \boldsymbol{\epsilon}_\phi(\mathbf{z}_t; \mathbf{c}, t) - \boldsymbol{\epsilon}_\phi(\mathbf{z}_t, t) \right) + \omega_{\text{con}} \left( \boldsymbol{\epsilon}_\theta(\mathbf{z}_t; \mathbf{c}, t) - \boldsymbol{\epsilon}_\phi(\mathbf{z}_t; \mathbf{c}, t) \right), \quad (13)$$

at each timestep $t$. Note that for fixed $\omega_{\text{text}}$, the fidelity to the reference images increases if using higher $\omega_{\text{con}}$, while this comes at the cost of losing prompt fidelity. On the other hand, one can improve prompt fidelity by using small $\omega_{\text{con}}$. We show that varying $\omega_{\text{con}}$ controls the tradeoff between fidelity to the reference and image-text alignment (*e.g.*, see Fig. 5). Note that a similar sampling method was introduced in [11], but for transformer-based T2I models [7]. While they do not use regularized fine-tuning objectives, the consistency guidance scheme is still valid. Thus, the consistency guidance could be implemented in any fine-tuned model, while we show that it is more effective when combined with a DCO fine-tuned model (*e.g.*, Fig. 5).

## 4.4 Prompt Construction for Reference Images

An important (yet often overlooked) part of the T2I personalization process is the prompt construction of reference images. Recall that Ruiz et al. [10] have proposed the use of a compact prompt in the form of "a photo of [V] [class]" with a rare token identifier [V]. However, we find that the use of compact caption is prone to learning distractors, such as a background or a style, as part of the fine-tuned model, as illustrated in Fig. 2.

**Comprehensive caption.** Instead, we propose to provide a comprehensive and visually grounded caption that not only describes the subject but also details other visual attributes, backgrounds, and styles of reference images. In Fig. 2, we show an example of a comprehensive caption and compare the synthesized results that use a compact caption. We find that providing detailed descriptions of the undesirable attributes, *e.g.*, background, or style, helps anchor desirable attributes in reference images to corresponding texts, making it easier to separate between them. This method not only holds for subject customization, but also for style customization; we provide comprehensive descriptions of the subject so that the model distinguishes style from the subject. Note that the use of comprehensive caption has been considered in practice,[4] but has not been investigated from the lens of model shift and concept disentanglement. In our experiments, we use vision-language models such as GPT-4 [50] or LLaVA [51] (*e.g.*, see Fig. 11).

## 5 Experiments

We use Stable Diffusion XL [6] for our experiments. We conduct experiments on subject (Sec. 5.1), style (Sec. 5.2) personalization, and their combination (Sec. 5.3). Ablative studies are in Sec. 5.4.

### 5.1 Subject Personalization

**Experimental setup.** We conduct experiments on DreamBooth dataset [10], containing 30 subjects, with 4–6 images per subject. The examples of images and captions are in Fig. 11 in Appendix D.1.

---

[4]See this blog post as an example.

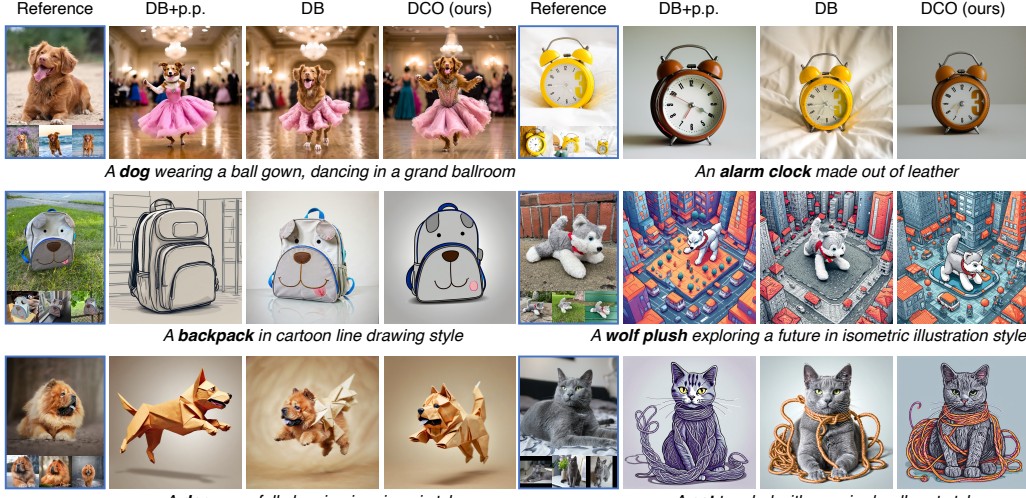

Figure 3: **Custom subject generation.** We show selected generations from DreamBooth (DB), DB with prior preservation (DB+p.p.), and ours (DCO) of custom subjects with varying attributes and styles guided by text prompts. While DB captures subjects well, it does not follow text prompt well. DB+p.p. shows better textual alignment, but falls short in subject fidelity. Ours show the best in both image-text alignment and subject fidelity. Best viewed in color, zoomed in on monitor.

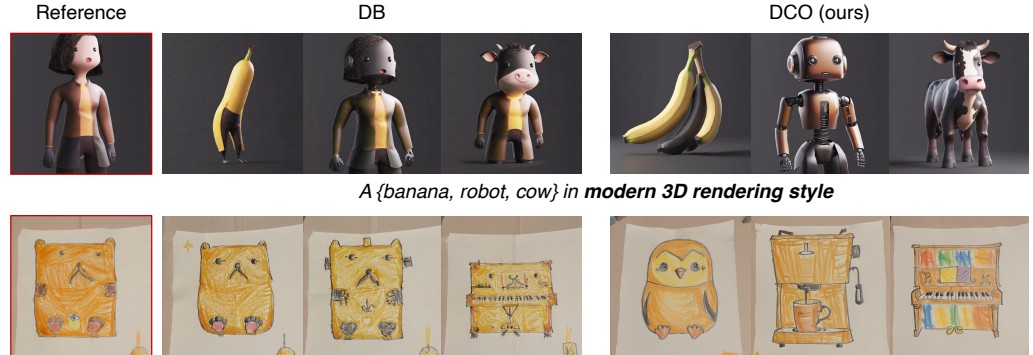

Figure 4: **Custom style generation.** We show selected generations from DreamBooth (DB) and ours (DCO) of custom styles with varying subjects. DB is prone to capturing undesirable attributes, resulting in generation of mixed concepts (*e.g.*, the girl's outfits in the first row, the dog in the second row), whereas DCO mitigates such a concept mixing. Best viewed in color, zoomed in on monitor.

For baselines, we consider the training with a regular diffusion loss, *i.e.*, DreamBooth (DB), and the one with prior preservation loss (DB+p.p.). Note that we additionally optimized textual embeddings, *i.e.*, textual inversion [9], which enhances subject fidelity for all baselines, though we omit for clear context. For all experiments, we fine-tune LoRA of rank 32 and textual embeddings using Adam [52] optimizer with learning rates of 5e-5 and 5e-4, respectively. We use constant $\beta_t$=1000 for DCO loss.

Following Sec. 4.4, we provide comprehensive and visually grounded caption that not only describes the subject but also details other visual attributes, backgrounds, and styles of reference images, for fine-tuning. Note that this is used for both baselines and ours. We observe that comprehensive captioning improves all the baselines, thus we omit the indications of its usage.

**Qualitative results.** Fig. 3 shows the qualitative comparison of our approach with DreamBooth (DB) and DreamBooth using prior preservation loss (DB+p.p.). We observe that our approach generates images of various visual attributes, *e.g.*, outfits and backgrounds, or changing the material, as well as into various styles, *e.g.*, in origami style or doodle art style. While DB changes the background, it lacks recontextualization in different outfits or styles, especially due to the overfitting to the photographic style. DB+p.p. is better than DB in prompt fidelity, but it often fails to preserve the subject identity.More qualitative comparisons are demonstrated in Fig. 17.

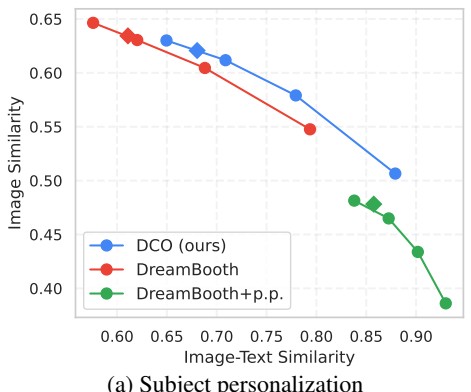
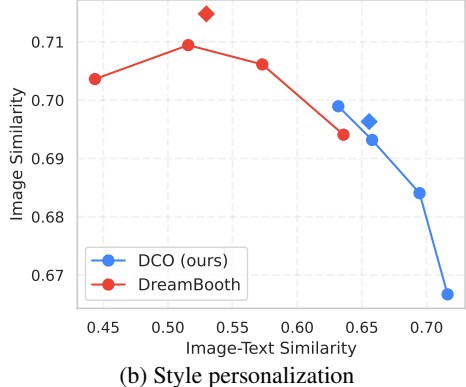

(a) Subject personalization  (b) Style personalization

Figure 5: **Quantitative results.** We plot Pareto curve between subject / style fidelity (image similarity) and prompt fidelity (image-text similarity) on (a) subject personalization and (b) style personalization tasks. Scores of each point are measured with consistency guidance sampling (dots and lines) of $\omega_{con} = 2.0, 3.0, 4.0, 5.0$, and conventional classifier-free guidance sampling (diamond). See Sec. 5.1 and Sec. 5.2 for experimental details, and Appendix B.1 for full comparison.

**Quantitative results.** For quantitative evaluation, we design two types of text prompts: subject customization, where we modify the attributes of the subject or its background, and subject stylization, where we change the visual style of a subject image. For each subject and type, we generate images from 50 text prompts with 2 random seeds. For evaluation metrics, we report the image similarity score using DINOv2 [53] and the image-text similarity score using SigLIP [54]. We also show the results using DINO [55] and CLIP [45] in Fig. 9, showing similar trends. See Appendix D for details.

Noting that this is a multi-objective problem (*i.e.*, maximizing image similarity and image-text similarity), we report the Pareto curve consisting of scores with varying consistency guidance scale values to show the tradeoff between two scores of each model, instead of reporting scores at one operating point. If two curves overlap, two methods would likely perform similarly and the difference is up to a change in the consistency guidance scale at sampling.

Averaged results are in Fig. 5a, and the results for each subject customization and stylization are in Fig. 8a and Fig. 8b, respectively. Compared to DreamBooth (DB;red), ours (DCO;blue) positions on the upper-right frontier in both image-text similarity and image similarity, demonstrating its superiority. Compared to DreamBooth with prior preservation loss (DB+p.p.; green), ours (blue) results in significantly improved image similarity, while being comparable in image-text similarity. Interestingly, it (green) does not push the frontier to the upper-right compared to the ones without it (red), but it shifts the operating point to the lower-right while lying on the seemingly similar Pareto frontier. This suggests that the use of prior preservation loss improves prompt fidelity at the cost of losing the subject consistency. In Appendix B.1, we further compare with various design choices for DreamBooth, *e.g.*, early stopping or lowering $\lambda_{prior}$ for prior preservation loss.

### 5.2 Style Personalization

**Experimental setup.** We experiment on style images from StyleDrop dataset [11]. The examples of style images and captions are in Fig. 12 in Appendix D.1. We fine-tune LoRA of rank 64 using Adam optimizer with a learning rate of 5e-5 and do not train textual embedding. In addition, we add an offset noise [56] of $0.1$ during training, which empirically helps learning the solid background color of style images. We use $\beta_t$=1000 for DCO loss, and compare with DreamBooth (DB).

**Qualitative results.** Fig. 4 shows qualitative comparisons between DreamBooth (DB) and ours (DCO). As seen in [21], DB captures the style of a reference image, yet it suffers from overfitting to the reference image, *e.g.*, the attributes of the subject in style reference appear in generated images. On the other hand, DCO generates images with consistent style without being entangled with contents in the reference images. We provide additional comparisons in Fig. 18.

**Quantitative results.** We choose 10 style images and generate stylized images using 190 text prompts excerpted from Parti prompts [5], following [11]. We generate 2 images per evaluation prompt, resulting in 380 images in total for each style. For evaluation metrics, we report the image similarity against the style reference image and image-text similarity scores using SigLIP [54].

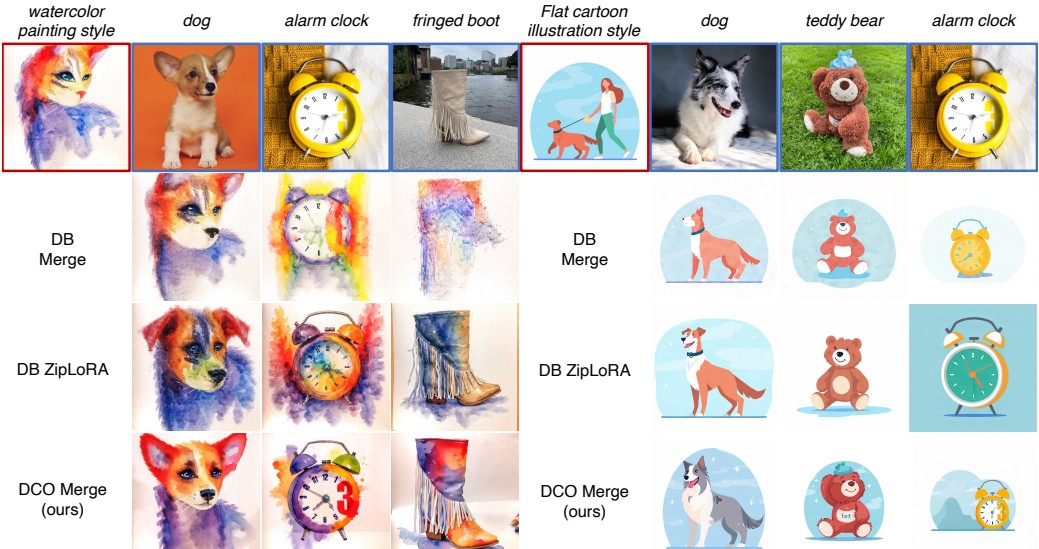

Figure 6: **My subject in my style generation.** We show generated images by merging subject and style LoRAs, each trained independently with DB (DB Merge) or DCO (DCO Merge). We also show results of ZipLoRA [21] using DB models (DB ZipLoRA). DB Merge struggles to generate high quality images. While DB ZipLoRA improves the quality, it less preserves fidelity. DCO Merge produces consistent images in both subject and style. Best viewed in color, zoomed in on monitor.

Fig. 5b shows results. While two models operate in somewhat disjoint regimes, we see that the curve of DCO (ours) is placed on the right of that of DB, showing improved image-text similarity. Nevertheless, as noted in [11], the image similarity for style personalization is particularly noisy, as the score is not only guided by the style but also by the unexpected appearance of the subject in the style reference image (*e.g.*, in Fig. 4) when the model overfits.

### 5.3 My Subject in My Style

**Experimental setup.** Following [11], we combine customized subject and style models to generate images of *my subject in my style*. Specifically, given two LoRAs $\Delta W_1$ and $\Delta W_2$ for subject and style, respectively, we use an arithmetic merge (Merge) [21], *i.e.*, $\Delta W = \tau_1 \Delta W_1 + \tau_2 \Delta W_2$ with coefficients $\tau_1 = \tau_2 = 1$. We use subject and style LoRAs from Sec. 5.1 and Sec. 5.2, respectively, for both baseline (DB) and our method (DCO). We also compare with ZipLoRA [21], which finds optimal coefficients $\tau_1$ and $\tau_2$ for each layer by jointly preserving the subject and style LoRAs. For ZipLoRA, we use DB fine-tuned subject and style models and follow the experimental setup in [21].

**Qualitative results.** In Fig. 6, we provide qualitative comparisons between our approach (DCO Merge), and baselines (DB Merge and DB ZipLoRA). As noticed in [21], DB Merge struggles to generate high-quality images when composing subject and style customized models. While DB ZipLoRA improves the image quality with fewer artifacts, it often loses subject or style consistency. Even using a simple arithmetic merge, DCO Merge (ours) generates images with high subject and style consistency. In addition, Fig. 1 and Fig. 20 in the appendix shows that DCO Merge successfully generates my subjects in my styles under various contexts, guided by text prompts.

**Quantitative results.** We use 30 subjects from DreamBooth [10] dataset and 10 style images from StyleDrop dataset [11] from Sec. 5.1 and Sec. 5.2, respectively. For each subject and style pair, we generate images of "A [subject] in [style]" and of various text prompts that change attributes, backgrounds, or actions (*e.g.*, in Fig. 20). We compute subject similarity scores (`Subject`) using DINO v2 [53], style similarity (`Style`), and image-text similarity (`Text`) scores using SigLIP [54].

Tab. 1 reports results. DCO Merge significantly outperforms DB Merge and DB ZipLoRA in subject similarity (0.462 vs. 0.386, 0.406) and image-text similarity (0.773 vs. 0.430, 0.729), while retaining competitive style similarity (0.651 vs. 0.672, 0.662). This aligns with our observation in Fig. 6.

Table 1: **Quantitative results of my subject in my style generation.** We report subject, style, and image-text similarity scores of DB Merge, DB ZipLoRA, and DCO Merge (ours).

|         | DB Merge | DB ZipLoRA | DCO Merge |
|---------|----------|------------|-----------|
| Subject | 0.386    | 0.406      | **0.462** |
| Style   | **0.672**| 0.662      | 0.651     |
| Text    | 0.430    | 0.729      | **0.773** |

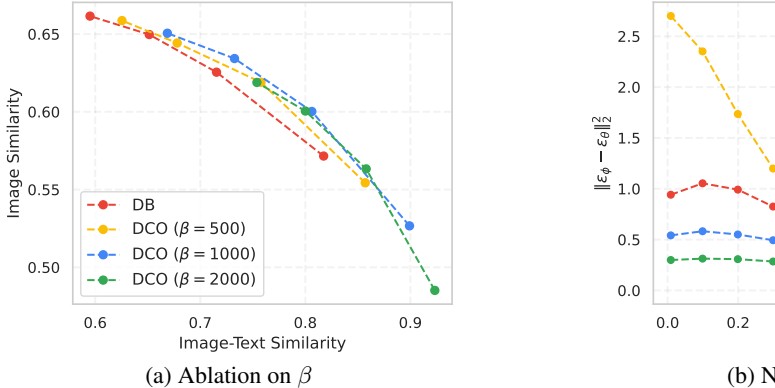

(a) Ablation on $\beta$          (b) Noise distance

Figure 7: **Ablation studies.** We conduct ablation studies on (a) regularization hyperparameter $\beta$, and (b) noise distance between pretrained model and fine-tuned model.

## 5.4 Ablation Study

**Regularization parameter.** We study the effect of the regularization parameter $\beta_t$ on the subject personalization task. We use 10 subjects from the DreamBooth dataset and conduct experiments with constant $\beta = \{500, 1000, 2000\}$. As in Fig 7a, all DCO models form a better Pareto frontier than the DB from Sec. 5.1. As $\beta$ becomes larger the curve tends to move lower right, implying the loss in subject identity for better image-text alignment. This is expected as $\beta$ controls the deviation between fine-tuned and pretrained models (*e.g.*, Eq. (6)). We find $\beta = 1000$ works well overall, though the optimal $\beta$ might vary across the reference dataset.

**Error analysis.** One of our insights is that DCO mitigates the shift in the model's generation distribution after fine-tuning. We verify this by computing the noise distance between pretrained and fine-tuned models on reference images and prompts, *i.e.*, $\|\epsilon_\theta(\mathbf{z}_t; \mathbf{c}, t) - \epsilon_\phi(\mathbf{z}_t; \mathbf{c}, t)\|_2^2$ at each timestep $t \in [0, 1]$. We simulate 100 random noises at each timestep and report the average value in Fig. 7b. We see a clear decrease in noise deviation from the pretrained model with DCO fine-tuning over DB. Moreover, as $\beta$ increases, the noise deviation gets further reduced as expected.

**1–shot personalization.** We demonstrate the capability of our method in 1–shot personalization. We refer to Appendix B.3 for experimental details and qualitative results are in Fig. 15 and Fig. 16.

## 6 Conclusion

We introduce a Direct Consistency Optimization (DCO), a novel training objective for robust low-shot fine-tuning of the T2I diffusion model. DCO learn new concepts by controlling the deviation from pretrained model, thus retaining capabilities of pretrained model. We show that DCO enhances the image-text alignment and sample quality of personalized T2I synthesis compared to regular diffusion fine-tuning. And together with consistency guidance sampling, DCO results in the most superior Pareto frontier in terms of image-text similarity and image similarity. Moreover, DCO induces easier composition of independently fine-tuned subject and style T2I models. Also, merging DCO models outperforms post-processing methods on regular fine-tuned models, *e.g.*, ZipLoRA [21]. We provide limitations and broader impact statements in Appendix E and Appendix F, respectively.

## Acknowledgement

We express our gratitude to Nataniel Ruiz and Viraj Shah for their help on ZipLoRA implementation and Meera Hahn for their feedback on the presentation of our paper. We also appreciate Jinyeop Kim and Younghyun Kim for their feedback and support on our manuscript and project page.

This work was supported by Institute of Information & communications Technology Planning & Evaluation (IITP) grant funded by the Korea government (MSIT) (No.RS-2019-II190075, Artificial Intelligence Graduate School Program(KAIST); No. RS-2024-00509279, Global AI Frontier Lab; No.RS-2021-II212068, Artificial Intelligence Innovation Hub; RS-2022-II220953, Self-directed AI Agents with Problem-solving Capability).

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

# Appendix

## A    Derivation

**Derivation of Eq. (9).** From Eq. (8), we have

$$f(\mathbf{z}_{0:1}, \mathbf{c}) = \beta \log \frac{p_\theta(\mathbf{z}_{0:1}|\mathbf{c})}{p_\phi(\mathbf{z}_{0:1}|\mathbf{c})} + \beta \log Z$$

$$= \beta \log \frac{p_\theta(\mathbf{z}_{0:1}|\mathbf{c})}{q(\mathbf{z}_{0:1}|\mathbf{x})} - \beta \log \frac{p_\phi(\mathbf{z}_{0:1}|\mathbf{c})}{q(\mathbf{z}_{0:1}|\mathbf{x})} + \beta \log Z,$$

where $Z$ is a normalization constant that does not depends on $\mathbf{x}$. Therefore, we have

$$\mathbb{E}_{q(\mathbf{z}_{0:1}|\mathbf{x})}[f(\mathbf{z}_{0:1}, \mathbf{c}) - \beta \log Z] = \mathbb{E}_{q(\mathbf{z}_{0:1}|\mathbf{x})}\left[\beta \log \frac{p_\theta(\mathbf{z}_{0:1}|\mathbf{c})}{q(\mathbf{z}_{0:1}|\mathbf{x})} - \beta \log \frac{p_\phi(\mathbf{z}_{0:1}|\mathbf{c})}{q(\mathbf{z}_{0:1}|\mathbf{x})}\right]$$

$$= \beta\big(D_{\mathrm{KL}}(q(\mathbf{z}_{0:1}|\mathbf{x}) \,\|\, p_\phi(\mathbf{z}_{0:1}|\mathbf{c})) - D_{\mathrm{KL}}(q(\mathbf{z}_{0:1}|\mathbf{x}) \,\|\, p_\theta(\mathbf{z}_{0:1}|\mathbf{c})))$$

$$= \beta\Delta(p_\theta, p_\phi; \mathbf{x}, \mathbf{c}),$$

which proves our claim.

**Derivation of Eq. (11).** Let us consider subseries $\mathbf{z}_{t:1}$ of $\mathbf{z}_{0:1}$ for $t \in (0, 1)$. Then, we define the deviation $\Delta_t(p_\theta, p_\phi)$ for $\mathbf{z}_{t:1}$ as follows:

$$\Delta_t(p_\theta, p_\phi) = D_{\mathrm{KL}}\big(q(\mathbf{z}_{t:1}|\mathbf{x}) \,\|\, p_\phi(\mathbf{z}_{t:1}|\mathbf{c})\big) - D_{\mathrm{KL}}\big(q(\mathbf{z}_{t:1}|\mathbf{x}) \,\|\, p_\theta(\mathbf{z}_{t:1}|\mathbf{c})\big). \tag{14}$$

It is known that one can express the KL divergence with $\epsilon$-prediction as follows (*e.g.*, see Appendix A.1 in [48]):

$$\frac{\mathrm{d}}{\mathrm{d}t} D_{\mathrm{KL}}\big(q(\mathbf{z}_{t:1}|\mathbf{x}) \,\|\, p_\theta(\mathbf{z}_{t:1}|\mathbf{c})\big) = \frac{1}{2}\lambda'_t \mathbb{E}_{\epsilon \sim \mathcal{N}(\mathbf{0},\mathbf{I})}\big[\|\epsilon_\theta(\mathbf{z}_t; \mathbf{c}, t) - \epsilon\|_2^2\big]. \tag{15}$$

Thus, the following holds:

$$\frac{\mathrm{d}}{\mathrm{d}t}\Delta_t(p_\theta, p_\phi; \mathbf{x}, \mathbf{c}) = \frac{1}{2}\lambda'_t \, \mathbb{E}_{\epsilon \sim \mathcal{N}(\mathbf{0},\mathbf{I})}\big[\|\epsilon_\phi(\mathbf{z}_t; \mathbf{c}, t) - \epsilon\|_2^2 - \|\epsilon_\theta(\mathbf{z}_t; \mathbf{c}, t) - \epsilon\|_2^2\big]. \tag{16}$$

Now it is straightforward to see that

$$\Delta(p_\theta, p_\phi) = \int_1^0 \frac{\mathrm{d}}{\mathrm{d}t}\Delta_t(p_\theta, p_\phi)\mathrm{d}t$$

$$= \int_1^0 \frac{1}{2}\lambda'_t \, \mathbb{E}_{\epsilon \sim \mathcal{N}(\mathbf{0},\mathbf{I})}\big[\|\epsilon_\phi(\mathbf{z}_t; \mathbf{c}, t) - \epsilon\|_2^2 - \|\epsilon_\theta(\mathbf{z}_t; \mathbf{c}, t) - \epsilon\|_2^2\big]\mathrm{d}t$$

$$= \frac{1}{2}\mathbb{E}_{t \sim \mathcal{U}(0,1), \epsilon \sim \mathcal{N}(0,1)}\big[\lambda'_t\big(\|\epsilon_\theta(\mathbf{z}_t; \mathbf{c}, t) - \epsilon\|_2^2 - \|\epsilon_\phi(\mathbf{z}_t; \mathbf{c}, t) - \epsilon\|_2^2\big)\big],$$

which proves Eq. (9). Then since the softplus function is convex, we use Jensen's inequality to show that

$$\mathcal{L}_\Delta(\theta; \mathbf{x}, \mathbf{c}) = -\log(\beta\Delta(p_\theta, p_\phi))$$

$$\leqslant \mathbb{E}_{t \sim \mathcal{U}(0,1), \epsilon \sim \mathcal{N}(\mathbf{0},\mathbf{I})}\left[-\log \sigma\left(\frac{\beta\lambda'_t}{2}\big(\|\epsilon_\theta(\mathbf{z}_t; \mathbf{c}, t) - \epsilon\|_2^2 - \|\epsilon_\phi(\mathbf{z}_t; \mathbf{c}, t) - \epsilon\|_2^2\big)\right)\right]$$

$$= \mathbb{E}_{t \sim \mathcal{U}(0,1), \epsilon \sim \mathcal{N}(\mathbf{0},\mathbf{I})}\big[-\log \sigma\big(-\beta_t\big(\|\epsilon_\theta(\mathbf{z}_t; \mathbf{c}, t) - \epsilon\|_2^2 - \|\epsilon_\phi(\mathbf{z}_t; \mathbf{c}, t) - \epsilon\|_2^2\big)\big)\big]$$

$$= \mathcal{L}_{\mathrm{DCO}}(\theta; \mathbf{x}, \mathbf{c}),$$

where $\beta_t = -\frac{1}{2}\beta\lambda'_t$.

Remark that one can consider weighted deviation $\Delta_w(p_\theta, p_\phi)$ that matches the general weighted $\epsilon$-prediction loss (*i.e.*, Eq. (1)), by using monotonic weighting function $w_t$ as follows:

$$\Delta_w(p_\theta, p_\phi) = \int_1^0 \frac{\mathrm{d}}{\mathrm{d}t}w_t\Delta_t(p_\theta, p_\phi)\mathrm{d}t$$

$$= \frac{1}{2}\mathbb{E}_{t \sim \mathcal{U}(0,1), \epsilon \sim \mathcal{N}(\mathbf{0},\mathbf{I})}\big[w_t\lambda'_t\big(\|\epsilon_\theta(\mathbf{z}_t; \mathbf{c}, t) - \epsilon\|_2^2 - \|\epsilon_\phi(\mathbf{z}_t; \mathbf{c}, t) - \epsilon\|_2^2\big)\big].$$

Note that for DDPM [40], *i.e.*, conventional $\epsilon$-prediction loss, it considers $-\frac{1}{2}w_t\lambda'_t = 1$.

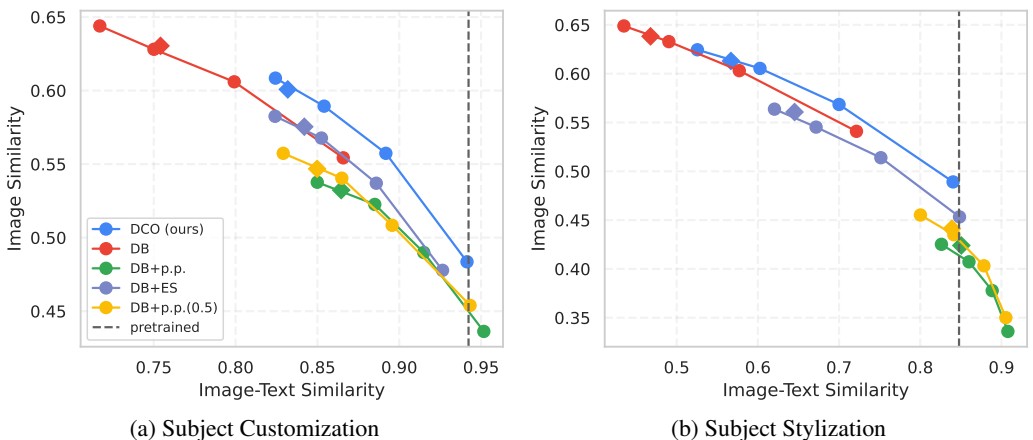

|                          |                          |
| :----------------------: | :----------------------: |
| (a) Subject Customization | (b) Subject Stylization |

Figure 8: **Quantitative results.** We plot image similarity and image-text similarity for each (a) subject customization and (b) subject stylization experiment. We use SigLIP [54] score for image-text similarity, and DINOv2 [53] score for image similarity. We plot the results of consistency guidance sampling (dots and solid lines), and conventional sampling (diamond). The reported reward guidance scales are $\omega_{con} \in \{2.0, 3.0, 4.0, 5.0\}$.

## B  Additional Experiments

### B.1  Full comparison

Here we provide full ablation studies that we have conducted in our experiments. For all 30 subjects in DreamBooth dataset, we follow the same experimental setup as in Sec. 5.1. We report the image similarity score using DINOv2 [53], and image-text similarity score using SigLIP [54].

**Ablation on early stopping.**  Since low-shot fine-tuning methods suffer from overfitting, it is a common practice to early stop the training. In Fig. 8b and Fig. 8b, we plot the results of DreamBooth with comprehensive caption with half of training steps (DB+ES). When compared to DB, it improves the image-text similarity ($0.842$ vs. $0.754$ for customization, $0.645$ vs. $0.468$ for stylization), while the image similarity significantly drops ($0.575$ vs. $0.754$ for customization, $0.630$ vs. $0.638$ for stylization). Notably, we remark that the frontier curve of DB+ES resides at the frontier of DB. Thus, early stopping does not improve the frontier.

**Ablation on prior preservation loss weight $\lambda_{prior}$.**  In Sec. 5.1, we show that using prior preservation loss often leads to loss of subject consistency. To further verify the effect of prior preservation loss, we vary the coefficient $\lambda_{prior}$ to be $0.5$ (DB+p.p. (0.5)). As shown in Fig. 8a and Fig. 8b, when compare to DB+p.p., using smaller $\lambda_{prior}$ improves image similarity ($0.547$ vs. $0.532$ for customization, $0.441$ vs. $0.424$ for stylization), while decreases the image-text similarity ($0.850$ vs. $0.864$ for customization, $0.839$ vs. $0.851$ for stylization). However, changing $\lambda_{prior}$ does not improve the frontier curve when using consistency guidance.

**Prior preservation loss vs. pretrained model.**  In Fig. 8a and Fig. 8b, we notice that DB with prior preservation loss (DB+p.p.) shows higher image-text similarity than pretrained model. This is in partly due to that the model is fine-tuned with class-specific prior dataset, which improves the prompt fidelity among the class. However, this does not necessarily improves the subject fidelity, and it indicates the large model shift with respect to pretrained model.

**Evaluation metrics.**  Following DreamBooth [10], in Fig. 9, we provide quantitative results using (a) CLIP [45] for image-text similarity (CLIP-T) and DINO [55] for image similarity (DINO), (b) CLIP for both image-text similarity and image similarity. We observe consistent trends to that which we used with SigLIP and DINOv2.

**User Evaluation.**  We conduct a user study over the results of our method (DCO), DreamBooth (DB), and DB with prior preservation loss (DB+p.p.) on subject personalization task. As in Section 5.1,

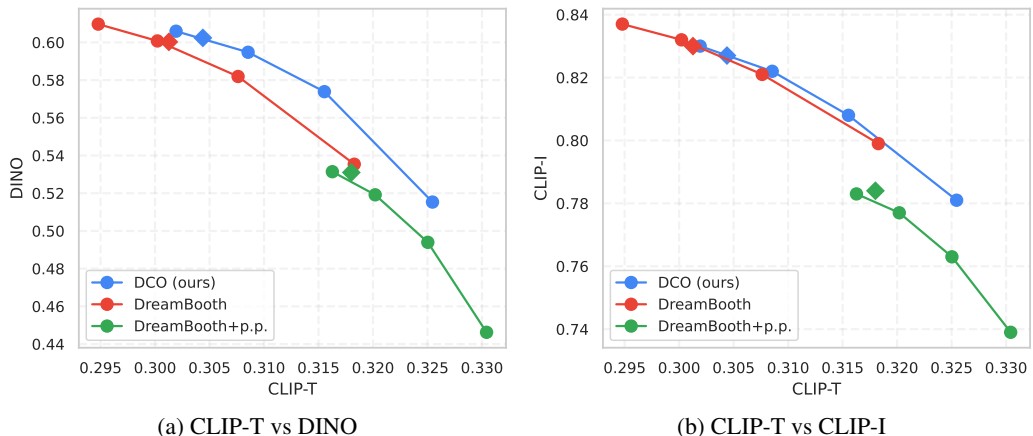

(a) CLIP-T vs DINO  (b) CLIP-T vs CLIP-I

Figure 9: **Quantitative results.** We plot image similarity and image-text similarity scores using (a) CLIP-T and DINO and (b) CLIP-T and CLIP-I. We plot the results of reward guidance sampling (dots and solid lines), and conventional sampling (diamond). The reported consistency guidance scales are $\omega_{\mathrm{con}} \in \{2.0, 3.0, 4.0, 5.0\}$.

Table 2: **User evaluation.** We asked the users to select better one given two images generated by DCO vs. baselines (DB and DB+p.p.) for each question: 1) subject fidelity, 2) image-text alignment, and 3) image quality. We report the percentage of judgements in favor of DCO over baselines.

| Win rate | vs. DB | vs. DB+p.p. |
|---|---|---|
| Subject fidelity | 55.1 % | 81.9 % |
| Prompt fidelity | 72.7 % | 58.0 % |
| Image quality | 70.6 % | 63.0 % |

we trained three models per subject with identical experimental setup (*i.e.*, all models are trained with comprehensive caption and textual inversion), and generate images with the same random seeds. Then we construct two binary comparison tasks (90 comparisons) to rank between DCO and each baseline method. For each pair, 12 participants were asked to choose the preferred image on three criteria with following questions:

- **Subject fidelity**: Which image most accurately reproduces the identity (*e.g.*, item type and details) of the reference item?
- **Prompt fidelity**: Which image most closely aligns with the given prompt?
- **Image quality**: Which image exhibits the highest quality (*e.g.*, overall visual appeal and clarity)?

Overall, we collect 1,080 answers per query for each comparison pair, resulting in a total 2,160 responses. The results are summarized in Table 2, where for each method, we report the percentage of judgements in our favor. Compared to DB, DCO shows better prompt fidelity while maintaining subject fidelity. Most users favored DCO for subject fidelity compared to DB+p.p., implying that DCO enables generating images with high subject and prompt fidelity.

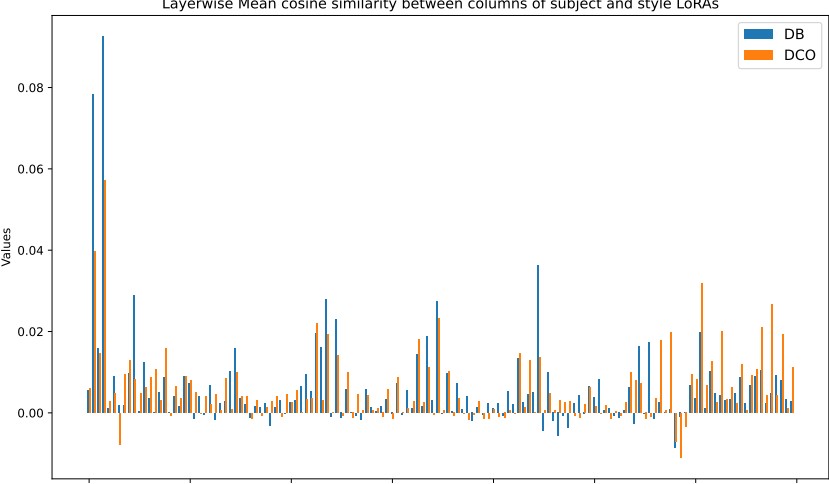

Figure 10: **Comparison on the alignment of DreamBooth and DCO fine-tuned subject and style LoRAs.** We compute the average cosine similarity of column layers between subject and style LoRAs fine-tuned with each DreamBooth (DB) [10] and our method (DCO). The x-axis denotes the component of each U-Net layer. The cosine similarity measures the alignment between two LoRAs, and high cosine similarity values are considered as the interference between them. Interestingly, we find that there is no obvious difference in the cosine similarity values between models trained with DCO and DB methods, while DCO fine-tuned models can be successfully combined with arithmetic merge to generate images of my subject in my style (*e.g.*, see Fig. 6 and Fig. 19). This may be in contrast with the findings of recent works [21, 57] and suggests further investigation on method to measure the compatibility between LoRA models.

## B.2 Effect of Consistency Guidance Scale.

We have shown that the user can control the subject fidelity and textual alignment by changing the consistency guidance scale. Yet, we remark that the optimal consistency guidance scale vary among the reference dataset, and even for the input prompt that the user give during inference. As shown in Fig. 14, the optimal guidance scale should be large (*e.g.*, 5.0) for the first row, while it should be low (*e.g.*, 2.0) for the third row. Also, the effect of consistency guidance scale might be subtle as in second row. In practice, choosing the best consistency guidance scale is up to the user's preference.

## B.3 1-shot personalization

Here we provide some qualitative examples that shows the capability of our method in learning subjects with single reference image. Specifically, we show the capability of personalization with synthetic images, generated by different T2I models such as pretrained SDXL [6] and DALLE–3 [8]. We follow the same setup as in Sec. 5.1 except that we fine-tune for 1000 steps. Remark that for while we have access to the prompts that we used for generation, we did not use this prompt for fine-tuning our model; the generated image might have more details (*e.g.*, backgrounds or attributes), or it may fail to capture all the prompts. Thus, we caption the images following Sec. 4.4.

Fig. 15 shows the qualitative results of DCO fine-tuning on images generated by SDXL. Remark that our method can synthesize images into various actions and styles, while preserving the subject consistency. Notably, it is possible to convert the photographs to other styles (*e.g.*, photo of a man to 2D animation style), and vice versa (*e.g.*, 3D animation style of pig into photography). Also, as shown in Fig. 16, our method is able to generate various actions, backgrounds, or styles.

## C Extended Related Work

**Training-free consistent image set generation.**    Several works have demonstrated the capability of consistent image set generation without fine-tuning T2I diffusion models [58, 59]. While these methods do not require fine-tuning and hence may be conceived more time-saving, they often

take longer time at generation. On the other hand, fine-tuning is a one-time cost and can be used for generation without additional cost. Also, training-free methods have difficulty in putting the same subject in different styles (as mentioned as limitation in [59]), while our approach is possible. Moreover, our method is able to combine style personalized model and subject personalized model. Lastly, our approach do not require any segmentation mask for subject personalization.

**Multi-concept personalization.** Given multiple fine-tuned T2I diffusion models (often using LoRAs), it is of great interest to combine them to generate a scene consists of multiple personalized subjects [60, 57], or generating custom subject in custom style [11, 21]. Those approaches often require post-optimization, *e.g.*, orthogonal adaptation of LoRA layers [57] or optimization of merger coefficients for composition of subject and style LoRAs [21]. Those methods hypothesize that the interference between subject and style LoRAs (which is measured by the average cosine similarity between LoRA layers), and aim to minimize the interference. Our method seeks for a better training of LoRA for T2I diffusion models and thus complementary to existing works. On the other hand, as shown in Fig. 10, we find that the cosine similarity values of DCO fine-tuned models are not necessarily smaller than those of DreamBooth fine-tuned models, while we do not observe significant interference during generation with arithmetically merged LoRAs trained with our DCO loss. This observation may suggest that we need another metric other than the cosine similarity based interference measure to evaluate the compatibility of LoRAs, which we leave as a future work.

# D Experimental detail

**Learning textual embeddings.** The rare token identifier [V], such as "sks", conveys undesirable semantics.[5] Thus, we opt to remove the rare token identifier and use the natural language captions by default. In addition, we learn textual embeddings [9] to add more flexibility in subject personalization without changing the semantics of pretrained model. Given a word or a phrase (*i.e.*, [class]) of interest, we insert new tokens and initialize them with textual embeddings of original ones. Then, newly inserted textual embeddings are optimized with diffusion models.

## D.1 Dataset

We use DreamBooth dataset [10] for subject personalization which contains 30 subjects, including pets and unique objects such as backpack, dogs, plushie, etc. We provide examples of image and comprehensive caption in Fig. 11 where the complete list of comprehensive captions are is available in the source code. A comprehensive caption encompasses not only the subject but also provides detailed information on visual attributes, backgrounds, and style. In contrast, a compact caption generally incorporated in model personalization [10, 9] focuses solely on the subject itself, as exemplified by "a photo of [class]". To generate comprehensive captions in practical scenarios, we initially employ LLaVA [51] to generate a description of the reference image. Subsequently, we filter out unnecessary details such as non-visual attributes and make further modifications. Similarily, we use 10 images from StyleDrop dataset [11] for style personalization, where examples are presented in Fig. 12.

**License.** The license for DreamBooth dataset can be found in here. Also, the attributes for style images can be found in StyleDrop [11] paper as well as here.

## D.2 Evaluation prompt

We construct two types of evaluation prompts; (1) subject customization, and (2) subject stylization. In evaluating subject customization, we provide the textual prompts that alter the attributes of the subject (*e.g.*, "cube-shaped") or its background (*e.g.*, "on the beach") following DreamBooth [10]. In evaluating subject stylization, we provide the textual prompts that stylizes the subject into different styles (*e.g.*, "in watercolor painting style", "in origami style"). For fine-grained evaluation, we construct different prompts for each object (*e.g.*, "clock", "robot toy") and subject (*e.g.*, "cat", "dog", "wolf plushie"). The examples of evaluation prompts for each category and type are in Table 3.

---

[5]See second and fourth rows of DB (baseline) in Fig. 17, where a dog is surrounded by guns and a monster toy is holding a gun. Note that this is aligned with the existing findings in community.

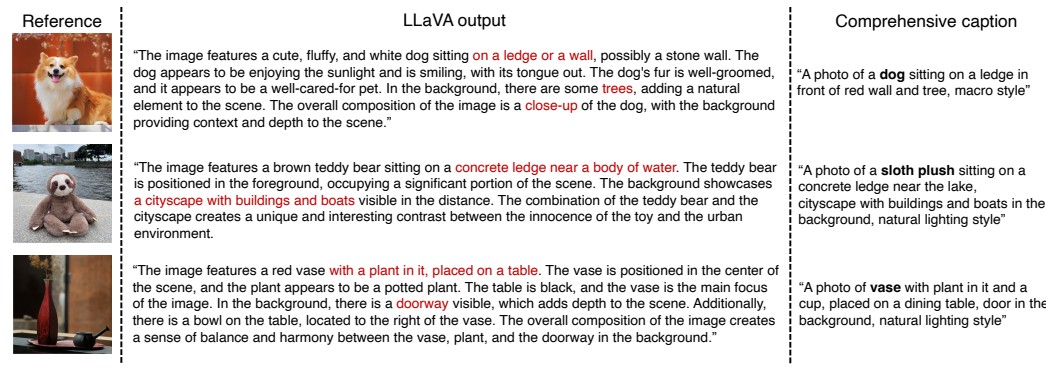

Figure 11: **Examples of comprehensive caption for subject images.** We provide examples of LLaVA [51] output and our comprehensive captions for each reference image. With help of LLaVA, we extract the visual attributes, backgrounds, and styles to construct comprehensive caption (*e.g.*, the texts marked in red in LLaVA output are used). The class tokens that are marked in bold (*e.g.*, dog, sloth plush, vase) are additionally learned with new textual embeddings initialized from the original one.

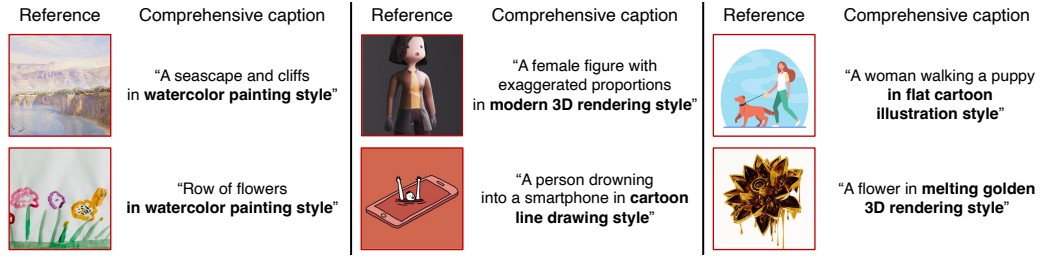

Figure 12: **Examples of comprehensive caption for style images.** We provide the examples of reference style images and comprehensive captions for style personalization experiment. To disentangle the subject and style in the image, we provide comprehensive description to the subject of the image. The texts marked in bold are used to generate image in custom style.

## D.3 Additional implementation detail

We use PyTorch and Huggingface Diffusers library[6] for our codebase. Each training is performed on a single A100 40GB GPU using a batch size of 1. We perform up to 2000 optimization steps. Note that our approach is robust to the length of training steps compared to the baseline (*i.e.*, DreamBooth), which often requires early stopping to prevent overfitting. We fine-tune LoRA of rank 32 for subject personalization and rank 64 for style personalization. For sampling, we use DDIM [41] scheduler with 50 steps, and use CFG guidance scale of 7.5 throughout experiments.

## D.4 Evaluation metric

To measure the image similarity, we use DINOv2 [53] score, which is given by the mean cosine similarity between the embeddings of reference images and synthesized images. To measure the image-text similarity, we use SigLIP [54] score, which is defined as

$$S_{IT}(\mathbf{x}, \mathbf{c}) = \frac{1}{1 + \exp\left(-f_{\text{img}}(\mathbf{x})^\top f_{\text{text}}(\mathbf{c}) + b\right)}$$

where $f_{\text{img}}$ and $f_{\text{text}}$ are $\ell_2$ normalized embeddings from image and text encoders, and $b$ is a bias term that is optimized during pretraining. We opt to use SigLIP score instead of CLIP score [45], as the range of CLIP score depends on the prompt and images, while SigLIP score provides a general score that is bounded on $[0, 1]$. For style personalization experiment, we measure image similarity using SigLIP image similarity, by computing the embeddings with SigLIP image encoder. While we desire

---

[6] https://github.com/huggingface/diffusers

Table 3: Examples of evaluation prompts used to synthesize images for each object and live subject category. Subject customization are prompts to generate novel views of photo-realistic images and subject stylization aims to alter style of the subject. {}'s are filled with the class token of the subject.

| Category\Type | Subject customization | Subject stylization |
|---|---|---|
| Object | "A photo of {} on the beach"
"A photo of {} with ribbons"
"A photo of cube-shaped {}"
"A photo of golden {}"
"A photo of {} made out of leathers"
"A photo of {} with a tree and autumn leaves in the background"
"A photo of {} on top of a white rug" | "A {} in sticker style"
"A {} in wooden sculpture"
"A {} in flat cartoon illustration style"
"A {} in pixel art style"
"A {} in wireframe 3D style"
"A {} in hygge style"
"A {} in geometric art style" |
| Subject | "A photo of {} wearing a spacesuit, planting a flag on the moon"
"A photo of {} as a firefighter, extinguishing a fire in a skyscraper"
"A photo of {} in a wetsuit, surfing a giant wave in the ocea"
"A photo of {} in Victorian attire, attending a tea party in an elegant garden"
"A photo of {} in a snowsuit, skiing down a steep mountain"
"A photo of {} as an explorer, navigating through an icy Arctic landscape"
"A photo of {} in an elegant masquerade mask at a Venetian ball" | "A {} playing a violin in sticker style."
"A {} carved as a knight in wooden sculpture"
"A {} piloting a hot air balloon in travel agency logo style"
"A {} constructed from abstract metal shapes in constructivism style"
"A {} on an epic quest in pixel art style"
"A {} designed as an intricate machine in blueprint style"
"A {} illustrated in an educational infographic style" |

high scores, these metrics are not perfect, *e.g.*, the image similarity can get 1.0 if the model overfits, otherwise, the image-text similarity can achieve high score if the model underfits. Thus, instead of reporting the scores from a single data point, we provide multiple data points of the same model while varying sampling parameters (*e.g.*, guidance scale values of consistency guidance sampling) and show the trends (*e.g.*, by showing the Pareto frontier).

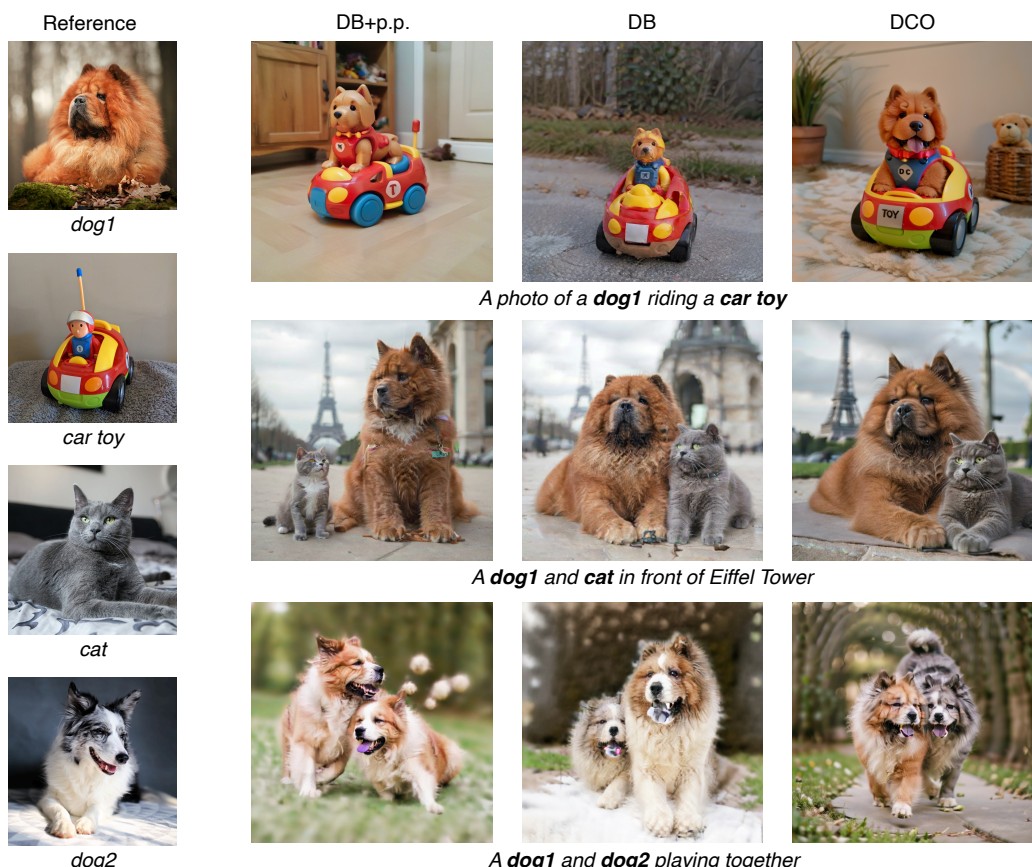

Figure 13: **Limitations.** We show the multi-subject composition results. While DCO (ours) do better than DreamBooth (DB) and DreamBooth with prior preservation loss (DB+p.p.) on multi-subject composition, the learned concepts are often mixed during generation (*e.g.*, the dogs are mixed in third row). We found this more frequently happens when learned concepts are similar to each other, *e.g.*, dog-dog composition is more challenging than dog-cat composition, and dog-cat composition is more challenging than dog-toy composition.

# E  Limitations

**Computational efficiency.**  Since our method leverages inference on pretrained model during training (*e.g.*, Algorithm 2) and inference (when using consistency guidance sampling) (*e.g.*, Eq. (13)), there exists a few extra computational burden compared to original DreamBooth fine-tuning or CFG sampling. Specifically, we measured the optimization and inference time per iteration. Compared to DreamBooth, our method (DCO) approximately takes $\times 1.3$ longer time in fine-tuning. Compared to CFG sampling, consistency guidance sampling requires $\times 2$ longer time in sampling. We believe that our work will motivate future studies on efficient fine-tuning to enhance scalability in practical scenarios.

**Multi-subject composition.**  We have shown that DCO effectively compose learned subject and style (*i.e.*, my subject in my style). When composing multiple subject, we found the performance is not stable. Fig 13 showcase some results in multi-subject compositional generation. We found that for subjects that are semantically distant, *e.g.*, dog-car toy or dog-cat composition, our method is able to generate multi-subject consistent images, while DreamBooth or DreamBooth+p.p. often fails to. However, when composing semantically similar subjects, *e.g.*, dog-dog composition, we found that our method fails as well. Specifically, the concepts are mixed during generation, which results in subject inconsistency. We believe this is partly due to the lack of model's capability in disentangling the newly learned concepts that are semantically similar. Thus, we believe using better pretrained model could ameliorate such issues.

# F  Broader Impact

This paper presents a method that enhances the performance of the personalization of T2I diffusion models. Similarly to other works, the technology for personalization of T2I diffusion models comes with benefits and pitfalls – the tool could be extremely effective for creative directors to efficiently generate new visual assets of various subjects or styles derived from existing private visual assets. Yet, the responsible use of the technology is required for protecting the ownership and copyright of individual assets.

Reference      CFG      $\omega_{\mathrm{con}} = 2.0$      $\omega_{\mathrm{con}} = 3.0$      $\omega_{\mathrm{con}} = 4.0$      $\omega_{\mathrm{con}} = 5.0$

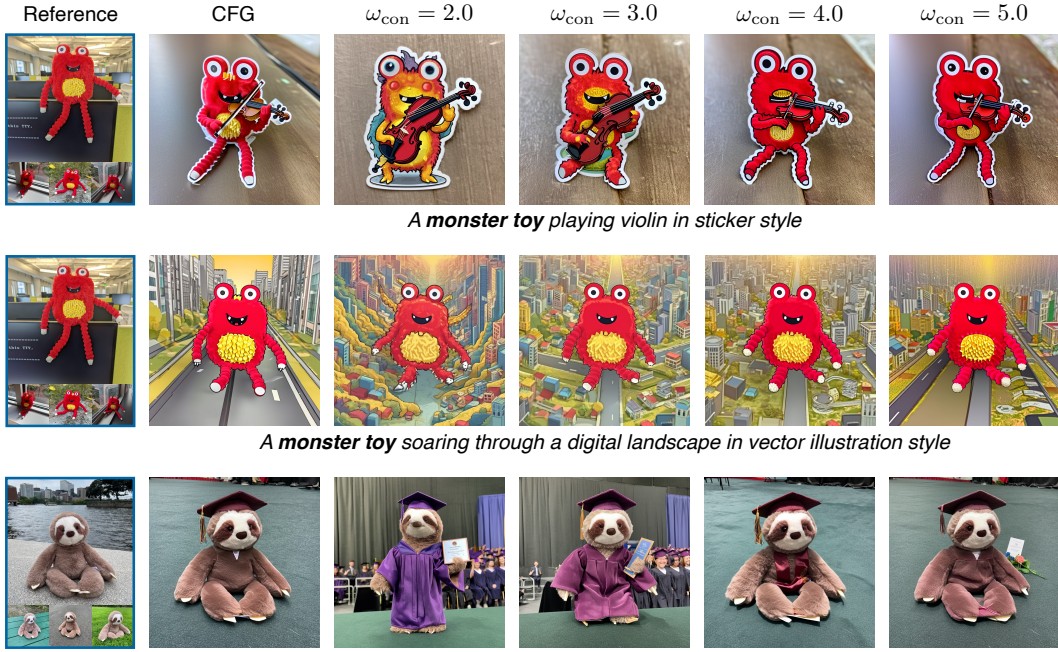

*A **monster toy** playing violin in sticker style*

*A **monster toy** soaring through a digital landscape in vector illustration style*

*A photo of **sloth plush** in a graduation gown, receiving a diploma on the stage*

Figure 14: **Effect of consistency guidance scale.** We show the effect of consistency guidance scale $\omega_{\mathrm{con}}$ by varying from 2.0 to 5.0. We also show the synthesized results using CFG. Note that the optimal choice of consistency guidance scale (in consideration of user's preference) might varies among reference dataset, or even input prompts.

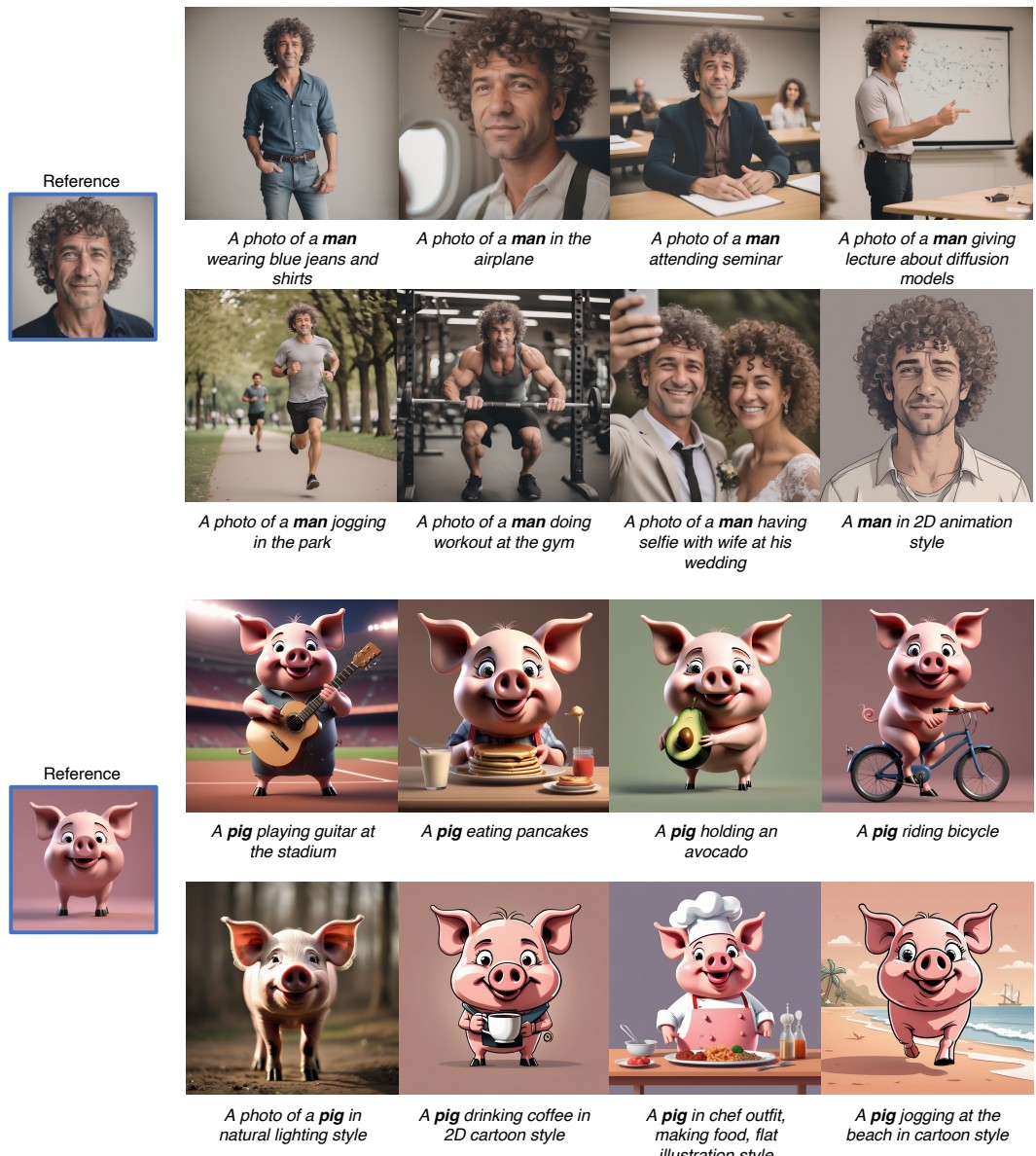

Figure 15: **1–shot personalization using synthetic images generatd by SDXL.** We show the capability of our method in 1–shot subject personalization using the images generated by pretrained SDXL models. For each reference image (man and pig), DCO fine-tuned T2I models can generate subjects with different actions and styles. The prompts that used to generate reference images were "a photo of a 50 years old man with curly hair" and "a 3D animation of happy pig", respectively, as used in [61].

Reference

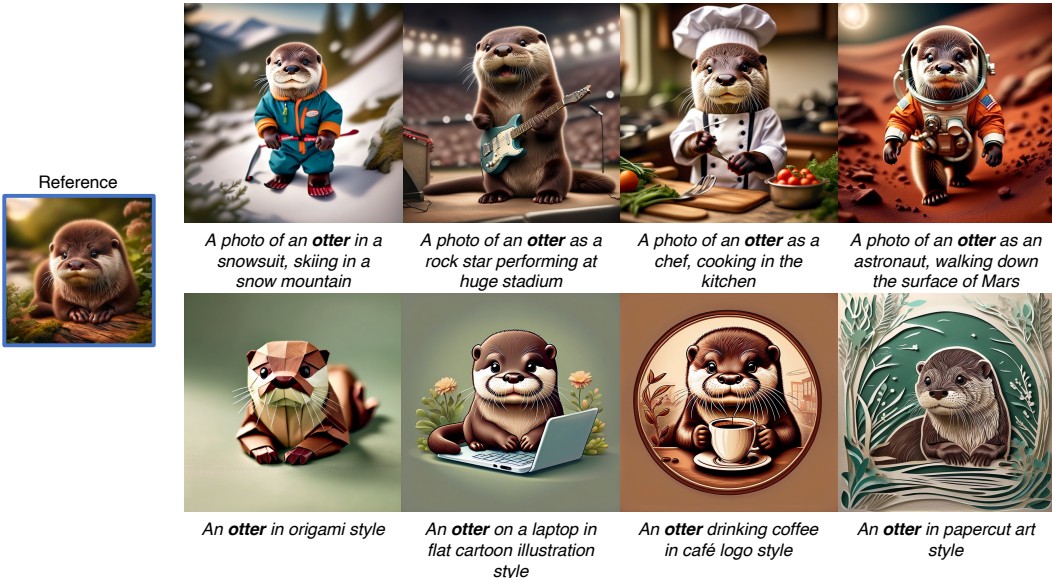

*A photo of an **otter** in a snowsuit, skiing in a snow mountain*

*A photo of an **otter** as a rock star performing at huge stadium*

*A photo of an **otter** as a chef, cooking in the kitchen*

*A photo of an **otter** as an astronaut, walking down the surface of Mars*

*An **otter** in origami style*

*An **otter** on a laptop in flat cartoon illustration style*

*An **otter** drinking coffee in café logo style*

*An **otter** in papercut art style*

Figure 16: **1–shot personalization using synthethic images generated by DALLE-3.** We show the capability of our method in 1–shot subject personalization using the images generated by DALLE–3 [8]. We asked DALLE-3 to generate a cute baby otter image. The comprehensive caption to fine-tune this image is "A closed-up photo of an <otter> on the top of wooden log, forest in the background". Our method is able to recontextualize the subject in the reference image with various text prompts depicting accessories, background, or style.

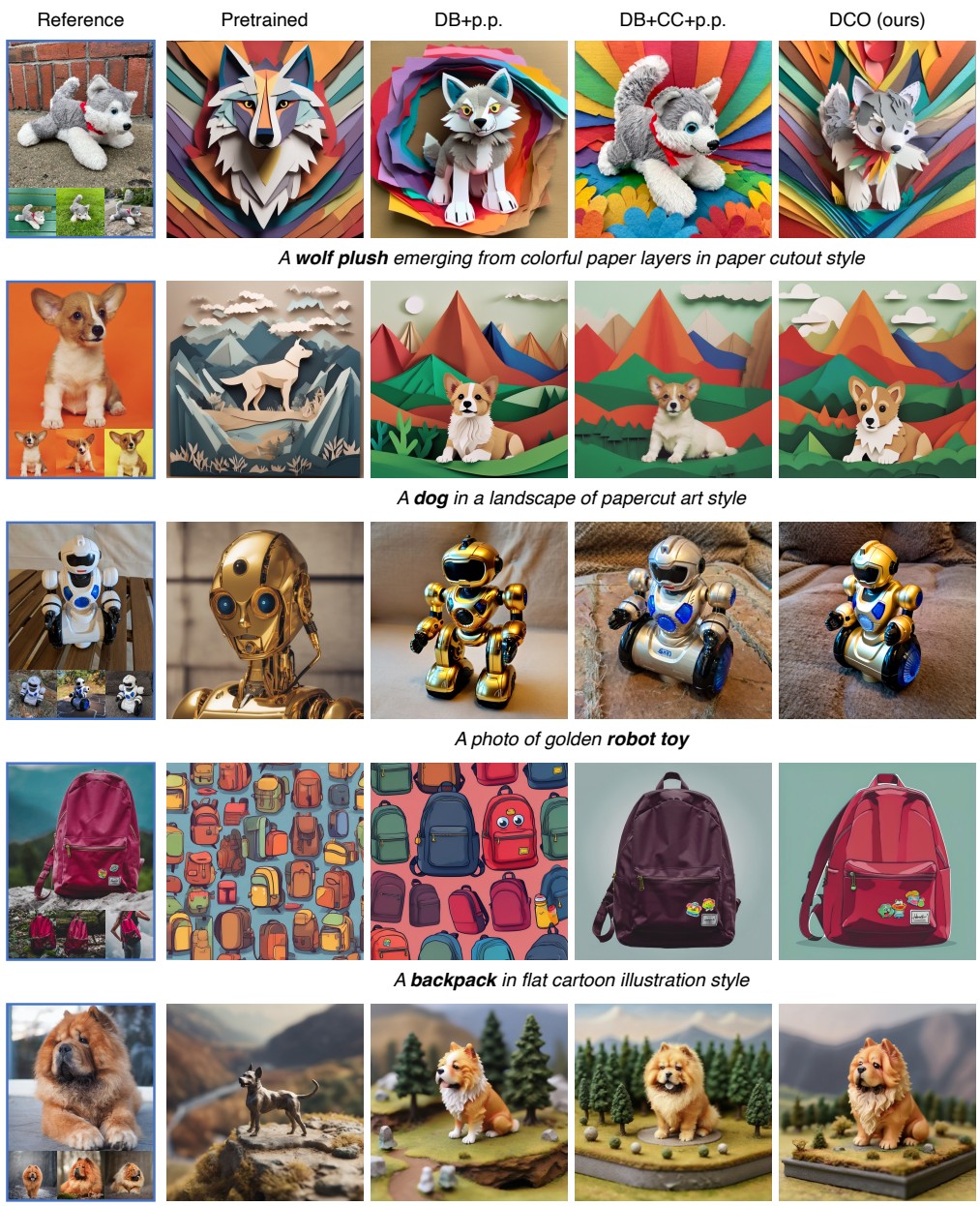

| Reference | Pretrained | DB+p.p. | DB+CC+p.p. | DCO (ours) |
|-----------|------------|---------|------------|------------|

*A **wolf plush** emerging from colorful paper layers in paper cutout style*

*A **dog** in a landscape of papercut art style*

*A photo of golden **robot toy***

*A **backpack** in flat cartoon illustration style*

*A **dog** as tiny figure in a grand landscape in miniature model style*

Figure 17: **Custom subject generation.** We compare our method (DCO) with pretrained model, DreamBooth (DB), and with prior preservation loss (DB+p.p.). Note that images in each row are generated using the same random seed. Our method is able to generate subject consistent images with various accessories, background or style, with better image-text alignment than baseline methods.

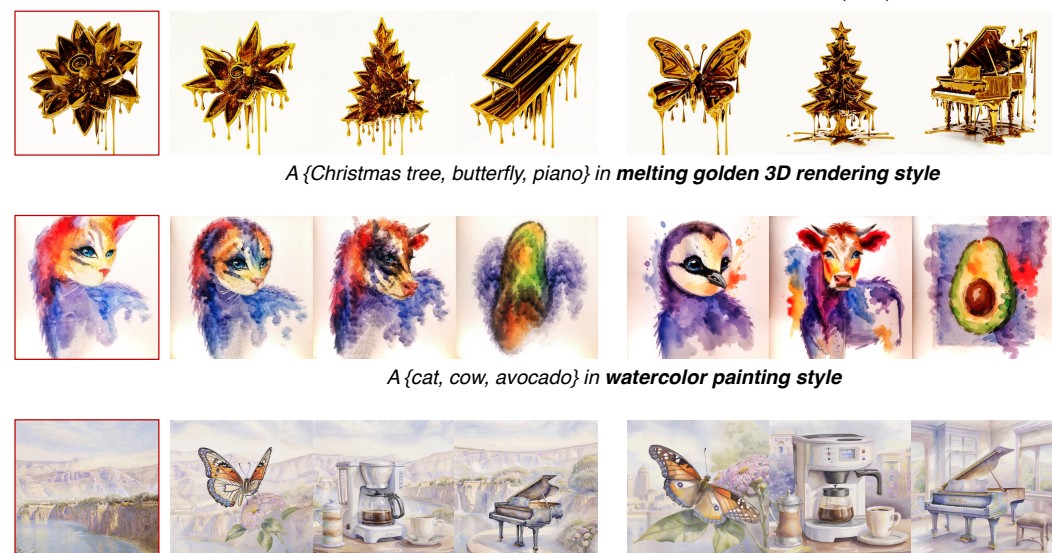

| Reference | DB | DCO (ours) |

*A {Christmas tree, butterfly, piano} in **melting golden 3D rendering style***

*A {cat, cow, avocado} in **watercolor painting style***

*A {butterfly, coffee maker, piano} in **watercolor painting style***

Figure 18: **Custom style generation.** Additional qualitative results on custom style generation. Our method (DCO) is able to generate style consistent image, while prior method, DreamBooth (DB), is prone to overfitting. For example, in the first row, the leaves of flower is inherited to butterfly, Christmas tree, and piano in DB, while our methods disentangle such attributes in generation. Those are also shown in second and third rows.

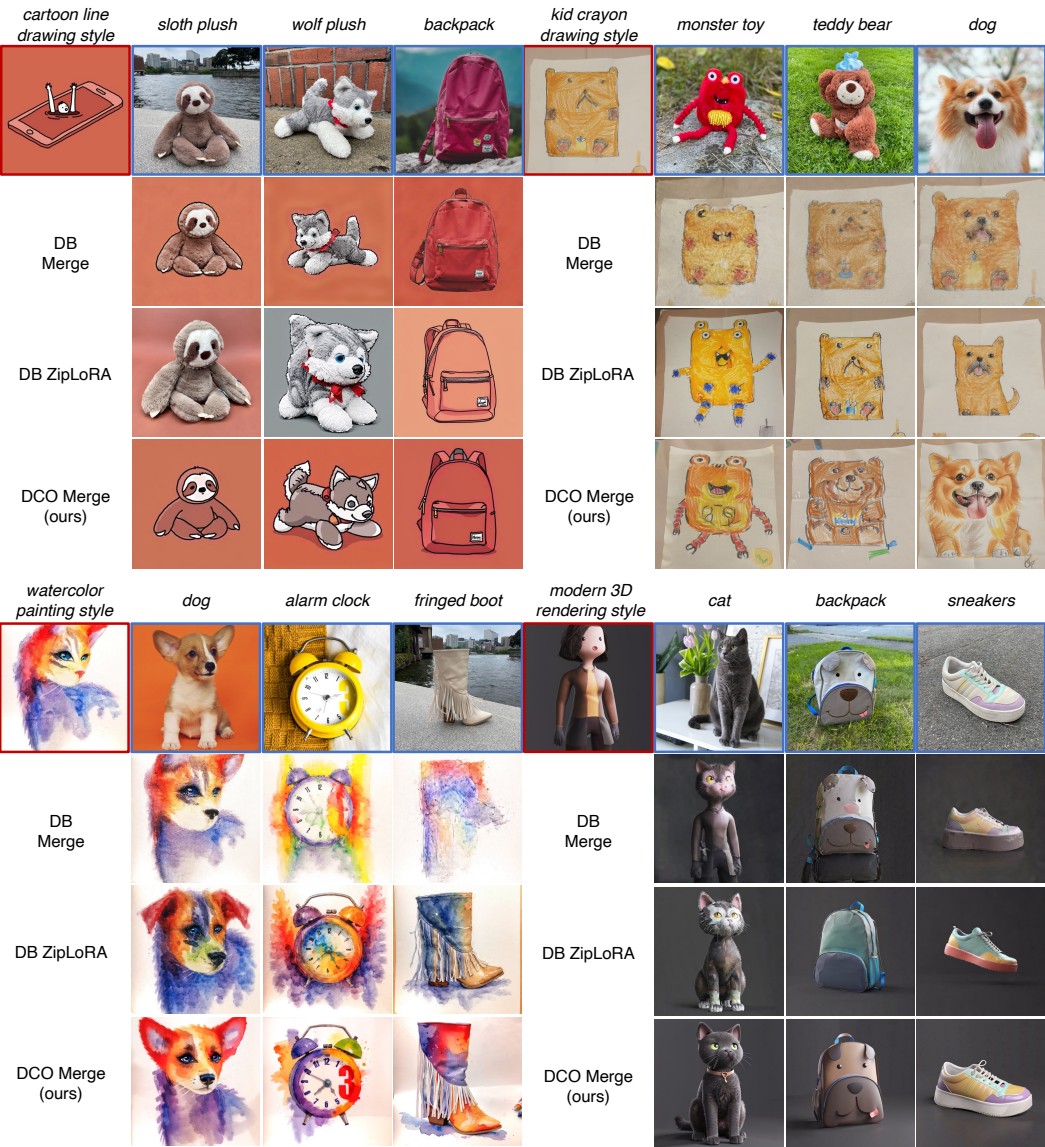

Figure 19: **My subject in my style generation.** Additional results are shown. Our method (DCO Merge) generates an image that maintains subject and style consistency without any post-processing. On the other hand, DreamBooth Merge (DB Merge) shows inferior results as is either overfitted to subject (*e.g.*, sloth plush, wolf plush, backpack are not in cartoon line drawing style), or styles (*e.g.*, monster toy, teddy bear, dog do not appear in kid crayon drawing style). Meanwhile, ZipLoRA shows better results than DB Merge, yet it often loses the subject or style fidelity.

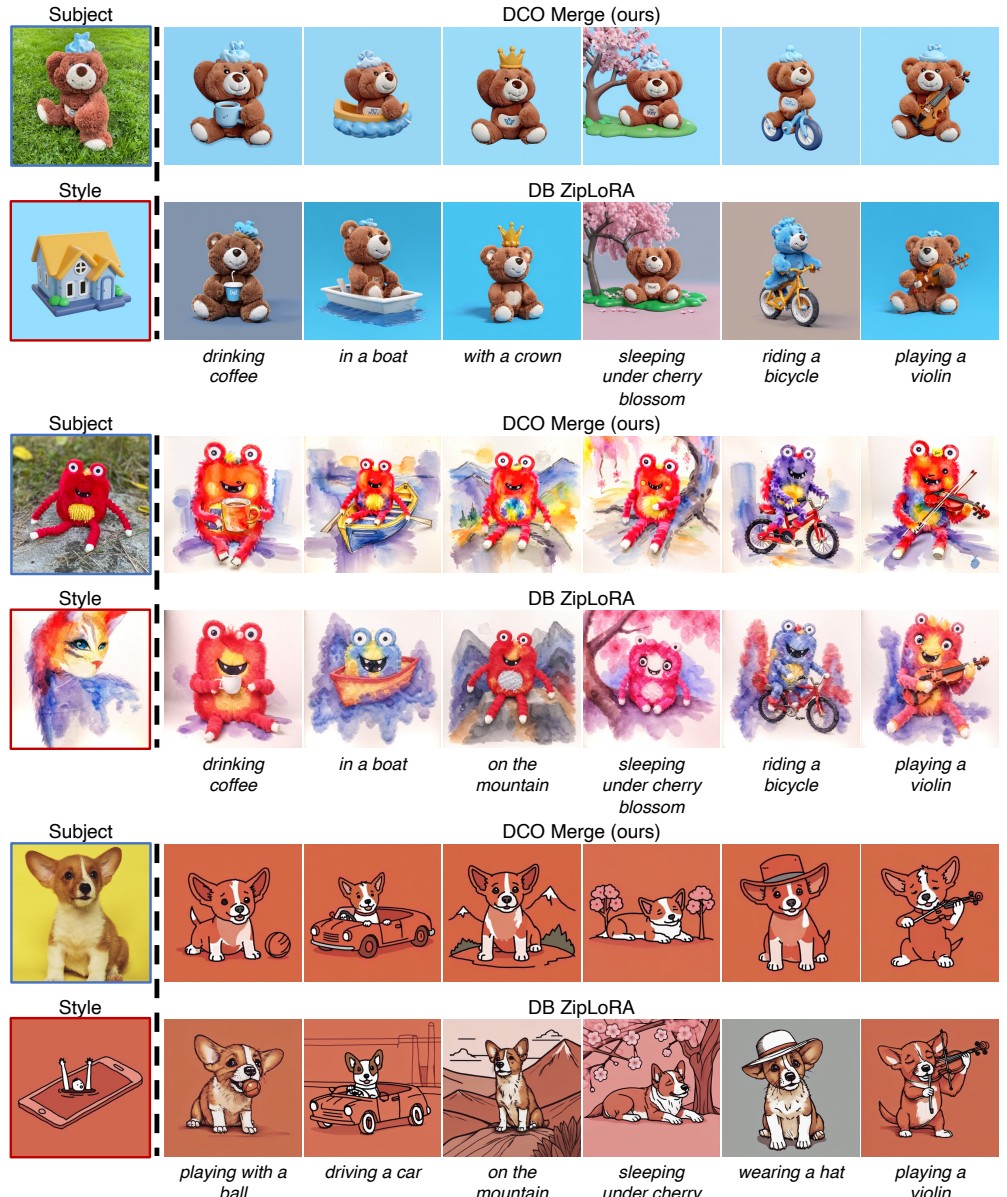

Figure 20: **Text-compositional my subject in my style generation.** We show more qualitative results for *my subject in my style* generation that compare arithmetic merge of DCO fine-tuned models (DCO Merge) and ZipLoRA [21] on DB fine-tuned models. DCO Merge generates images consistent to both subject and style without further post-optimization, while ZipLoRA often misses the subject or style consistency (*e.g.*, teddy bear and monster toy are changed, and the style of the third example is not as consistent).

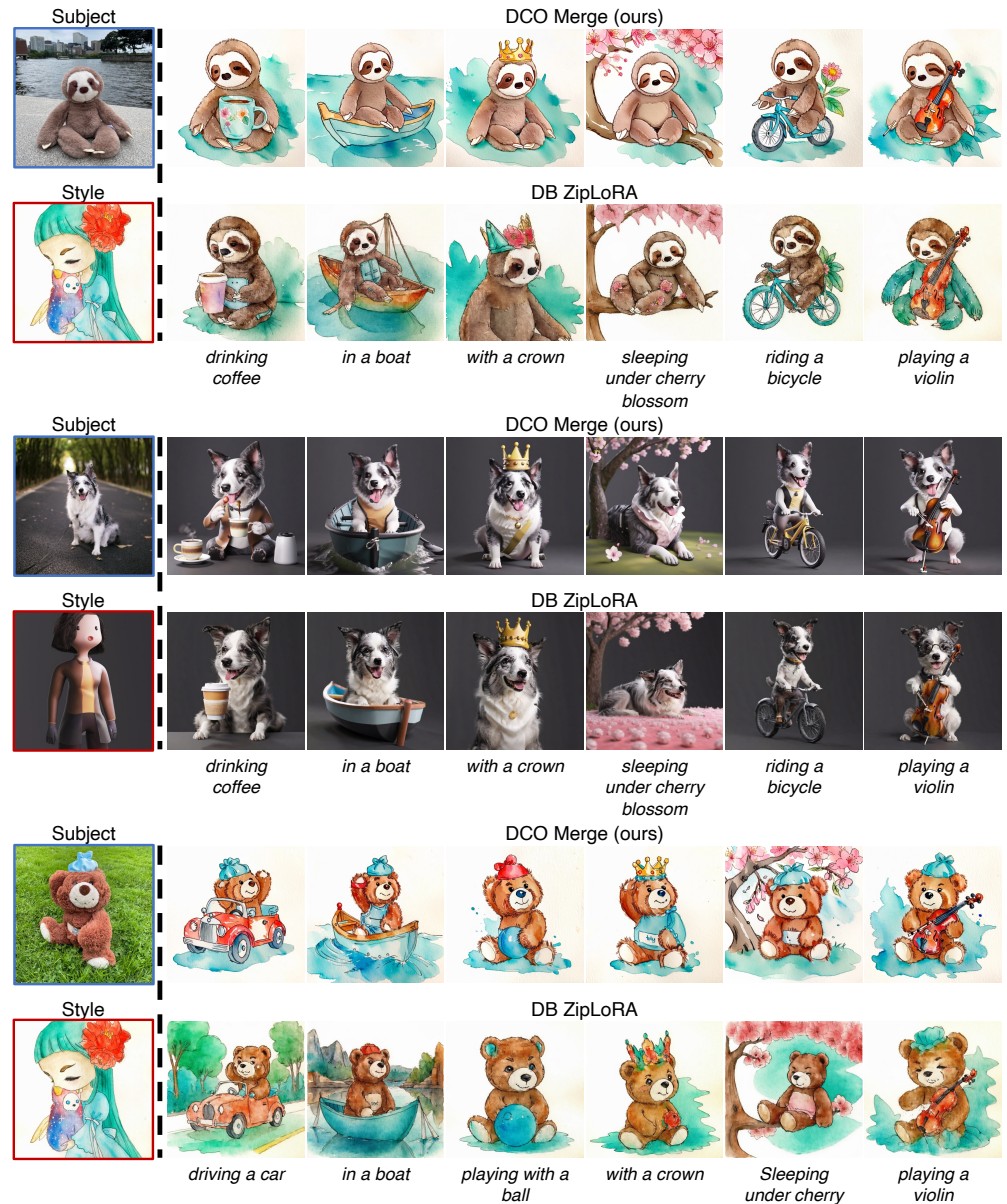

Figure 20: **Text-compositional my subject in my style generation. (continued)**

