# OpenReview forum: "Direct Consistency Optimization for Robust Customization of Text-to-Image Diffusion models"
_NeurIPS.cc/2024/Conference — NeurIPS 2024 poster_

### Official Review · Reviewer_XWYq · 2024-06-24

**Soundness:** 3
**Presentation:** 3
**Contribution:** 3
**Rating:** 6
**Confidence:** 5

**Summary:**

The paper introduces a novel training framework and novel sampling technique to enhance the state-of-the-art personalized and style-preserving adaptation of text-to-image diffusion models in the low-shot fine-tuning regime.

**Strengths:**

* The training objective introduced in the paper is implicitly incentivized to learn a reward for the model for better consistency.
* The decoupling of classifier-free guidance to include the learned consistency preserving model enhances image similarity with the reference image, thereby also improving image fidelity.

**Weaknesses:**

* DCO is computationally expensive since the proposed sampling scheme makes use of two networks during inference. However, the authors also mention it explicitly in the limitations section. But it would be nice to be forthright about it in the paper's main text.
* Multiple references to existing works are missing. Examples include SVDiff [1], Custom Diffusion [2], IP-Adapters [3], and, DiffuseKrona [4] (for controllable personalized generation), and MaPO [5] (reward fine-tuning through alignment). It also didn't include references to works that successfully applied LoRA in the context of adapting text-to-image models [6], [7], [8].
* The paper's main text reads incomplete without the minimum details on the datasets being used for training and evaluation. For example, in lines 219 - 224, the reader doesn't know what reference images were used. Similarly, in lines 251 - 254, there is no mention of Appendix D, which sheds light on more details. This is also the case for the text accompanying Table 1 (Lines 280 - 287).
* While reporting the results of Figure 4, it would have been better to talk about $\omega_{text}$ as well because that substantially impacts the generation quality. From the accompanying text, the reader doesn't know how it was varied or if it was kept fixed.

## References

[1] SVDiff: Compact Parameter Space for Diffusion Fine-Tuning, https://arxiv.org/abs/2303.11305.

[2] Multi-Concept Customization of Text-to-Image Diffusion, https://arxiv.org/abs/2212.04488.

[3] IP-Adapter: Text Compatible Image Prompt Adapter for Text-to-Image Diffusion Models, https://arxiv.org/abs/2308.06721.

[4] DiffuseKronA: A Parameter Efficient Fine-tuning Method for Personalized Diffusion Models, https://arxiv.org/abs/2402.17412.

[5] Margin-aware Preference Optimization for Aligning Diffusion Models without Reference, https://arxiv.org/abs/2406.06424.

[6] Implicit Style-Content Separation using B-LoRA, https://arxiv.org/abs/2403.14572.

[7] https://github.com/cloneofsimo/lora

[8] https://huggingface.co/blog/lora

**Questions:**

## Suggestions

* Figure 1.B is not obvious when conveying the limitations. Perhaps mention that the subject is not faithfully preserved (hat is missing) and the style is not faithfully preserved (blue background is missing) in the case of DreamBooth.
* Line 97 - 98: It would be better to also mention how training with classifier-free guidance is implemented in practice, i.e., we randomly make captions associated with images to null (caption dropout).
* Since the authors use LoRA for fine-tuning, I think the memory overhead could still be kept to a minimum if they enable and disable the adapter layers within the denoiser network. See how it's implemented [here](https://github.com/huggingface/diffusers/blob/3fca52022fe0ea9aaf0a0ea8a0fc13308bf69a9f/examples/research_projects/diffusion_dpo/train_diffusion_dpo_sdxl.py#L1019) and [here](https://github.com/huggingface/diffusers/blob/3fca52022fe0ea9aaf0a0ea8a0fc13308bf69a9f/examples/research_projects/diffusion_dpo/train_diffusion_dpo_sdxl.py#L1035). This probably should improve the memory requirements quite a bit. Could the authors verify this approach?
* It would be nice to enlist and cite the libraries that were used in the codebase. They do enable productive research and citing them goes a long way.
* Provide a loss landscape comparison when varying the amount of DCO $\beta$.
* Highlight the highest scores reported in Table 1.
* If both GPT-4 and LlaVA were used for captioning, how were the combined captions used during training?

## Questions

* If there's a connection between the guidance scheme introduced in InstructPix2Pix [1] and the one introduced in DCO, it would be nice to have that discussed.
* Line 204: It's not clear what do the authors mean by "omitting.
* For the DreamBooth and DreamBooth + PP experiments, do the authors use LoRA? If so, the text is not clear.
* Was consistency sampling applied for the DreamBooth experiments?
* Perhaps refer to the combination of LoRA DreamBooth + Textual Inversion as "pivotal tuning" as commonly referred in the community [2]?
* With a higher value of $\omega_{con}$, I would assume that the image similarity would improve. But with increasing $\omega_{con}$, it seems to be dropping off as per Figure 4. Or am I looking at it wrongly?
* Did the authors experiment with other merging techniques geared toward LoRAs [3]?

## Nits

* "In specific" could be replaced with "specifically".
* Line 46: "... policy optimization problem [19, 20], which amounts _for_ the consistency ..."
* Line 75: "_To the best of our knowledge,_ this paper is the first to have studied consistency as a reward function for T2I personalization."
* Line 83: Shouldn't $\lambda_{1}$ be $\lambda_{t}$? Similar comments for $\lambda_{0}$.

## References

[1] InstructPix2Pix: Learning to Follow Image Editing Instructions, https://arxiv.org/abs/2211.09800

[2] https://replicate.com/cloneofsimo/lora-training/readme

[3] https://huggingface.co/blog/peft_merging

**Limitations:**

The authors are clear about limitations. Perhaps mentioning them a bit from the main text would be more sensible.

---

> ### Author Rebuttal · Authors · 2024-08-07
>
> Dear reviewer XWYq,
>
> Thank you for valuable comments and suggestions in reviewing our work. We address each of your questions and concerns individually as follows.
>
> ---
> ### [W1] Regarding computational cost
> We will move the limitation into the main text in our final revision.
>
> ---
> ### [W2] References
> Thank you for the relevant work. While several papers have proposed different adapter designs (e.g., SVDiff, DiffuseKrona, B-LoRA) for improving personalized T2I synthesis, we introduce a new fine-tuning objective that is independent of adapter design. We appreciate the mention of the MaPO paper, which was unavailable at our submission time but will be included in our final manuscript as it provides a different fine-tuning objective for T2I diffusion models.
>
> ---
> ### [W3] Experimental dataset
> We elaborated the experimental dataset for each subject and style personalization experiment in line 200-211 and line 241-245, respectively. We will include details on the dataset for each figure and table in our final manuscript.
>
> ---
> ### [W4] The value of w_text
> $w_{text}$ is kept with 7.5 throughout the experiments (see Appendix D.4). We will revise our draft to include this information in figures.
>
> ---
> ### [S1] Regarding Figure 1
> We will add details to the captions of Figure 1 to better deliver the limitations of previous works and demonstrate effectiveness of our method.
>
> ---
> ### [S2] Details on classifier-free guidance
> We will add technical details of classifier-free guidance.
>
> ---
> ### [S3] Technical implementation
> Following your suggestion, we compare the training memory consumptions of suggested technical implementation and our implementation. Note that the suggested method use
> ```
> unet.disable_adapters()
> unet.enable_adapters()
> ```
> to compute loss from pretrained models. Originally, we used ‘cross_attention_kwargs’ to control the LoRA scale. For each method, we measure the allocated and reserved memory. All experiments were conducted on PyTorch v2.3.0, diffusers v0.27.2 on 1x A100 40GB GPU. The results are shown in the table below.
>
> |           | Mem. allocated | Mem. reserved | Max mem. allocated | Max mem. reserved |
> |-----------|----------------|---------------|--------------------|-------------------|
> | Ours      |   7812.78 MB   |  16653.48 MB  |     16021.28 MB    |    16653.48 MB    |
> | Suggested | 7812.78 MB     | 16647.19 MB   | 15933.40 MB        | 16647.19 MB       |
>
>
> The table shows that the suggested method saves more memory than our implementation. We will update technical implementation to our code, and release it with our final manuscript.
>
> ---
> ### [S4] Cite libraries
> We will add citations to the libraries that we used in our experiments.
>
> ---
> ### [S5] Loss landscape of DCO with different $\beta$
> We provide the loss landscapes by using different values of $\beta$ in Figure B in our attached file. For a small value of $\beta$, the regularization strength is small, thus the loss results in higher variance. For a large value of $\beta$, the regularization is stronger, which results in smaller variance.
>
> ---
> ### [S6] Highlight scores
> We will highlight the scores in Table 1.
>
> ---
> ### [S7] Training prompts
> Training prompts are manually engineered using vision language models like LLaVa or ChatGPT because their generated captions are often too long for SDXL text encoders. A more effective approach could be to use captioning models (e.g., in DALL-E 3) to generate prompts for personalized images similar to the pretraining datasets.
>
> ---
> ### [Q1] Consistency guidance
> InstructPix2Pix takes additional input (i.e., source image) to control the similarity of the edited image with respect to the source image. On the other hand, consistency guidance controls the fidelity to the reference image without any additional input. We will add discussion in our final revision.
>
> ---
> ### [Q2] Context of ‘omitting’
> To clarify, we omit the usage of textual inversion technique and comprehensive captioning, which we used for all baseline experiments for fair comparison.
>
> ---
> ### [Q3] Usage of LoRA
> As we mentioned in our paper, we used LoRA for all experiments.
>
> ---
> ### [Q4] Consistency sampling for DreamBooth
> For qualitative results, we show the results by using classifier-free guidance (CFG) sampling of fine-tuned models. In our quantitative results, we provide results of using both CFG sampling (diamond) and consistency guidance sampling (dots with solid lines) for all baselines including DB and DB+p.p.
>
> ---
> ### [Q5] Reference for pivotal tuning
> We will add citation of pivotal tuning for the usage of textual inversion and LoRA fine-tuning.
>
> ---
> ### [Q6] Regarding $w_{\text{con}}$
> Yes, as $w_{con}$ increases, the image similarities become higher. In Figure 4, the image similarities increase (i.e., towards upper left) as $w_{con}$ increases from 2.0 to 5.0.
>
> ---
> ### [Q7] Different merging method experiments
> Following your suggestion, we conduct experiments on comparing different merging methods. Specifically, we apply SVD based LoRA merging (i.e., TIES_SVD) [1]. As shown in Figure C, SVD merging improves quality for both DB and DCO compared to direct merging. We observe that LoRA trained with DCO shows better performance, particularly in following complicated prompts while maintaining identity. We will add this to our final revision.
>
> [1] Yadav, Prateek, et al. “Ties-merging: Resolving interference when merging models.” NeurIPS 2023.
>
> ---
> [N1-3] Thanks for pointing this out. We will revise in our final manuscript. \
> [N4] Note that $\lambda_1$ is the maximum log-SNR for the noise scheduling function, which should be large enough so that the distribution of $z_1$ follows pure random Gaussian noise. Conversely, $\lambda_0$ is the minimum log-SNR for the noise scheduling function, which should be small enough so that the distribution of $z_0$ matches the data distribution.

---

> > ### Comment · Reviewer_XWYq · 2024-08-08
> >
> > Thank you for comprehensively addressing my comments and running those experiments. Hope they were not too much trouble. I only have minor comments. Furthermore, I have raised my score to 6 from 5 after looking at your comments and rebuttal. All the best!
> >
> > [W2] I still think they at least deserve a mention. All of them target customization of T2I diffusion models at their very core. Maybe pick the ones that you think are most relevant and aligned.
> >
> > [S3] Glad it helped.
> >
> > [S5] Would it be included in the appendix of the main draft? I think it will be helpful for practitioners as it gives an idea about convergence.
> >
> > [S7] Thanks for the clarification. Perhaps this could be clarified in the main text.

---

> > > ### Author Response · Authors · 2024-08-08
> > >
> > > Thank you for your positive feedback and for raising the score. We appreciate your thoughtful comments and are glad the additional experiments were helpful.
> > >
> > > Regarding [W2], we will ensure to mention and discuss the relevant works that align with our study, as they indeed target customization of T2I diffusion models at their core.
> > >
> > > For [S3], we are pleased that the clarification was useful. We will update the technical implementation in our code release to reflect this.
> > >
> > > In response to [S5], we will include the training loss landscape in the appendix of the main draft, as we agree it will be beneficial for practitioners to understand convergence.
> > >
> > > For [S7], we will clarify this point further in the main text to enhance clarity.
> > >
> > > Thank you once again for your valuable feedback, which has significantly improved the quality of our work. If you have any further comments or suggestions, please do not hesitate to share them.

---

> > > > ### Comment · Reviewer_XWYq · 2024-08-09
> > > >
> > > > I acknowledge reading your comment, and I wish you all the best!

---

> > > > > ### Comment · Reviewer_ayXu · 2024-08-10
> > > > >
> > > > > I ack the rebuttal

---

### Official Review · Reviewer_ayXu · 2024-07-03

**Soundness:** 3
**Presentation:** 3
**Contribution:** 2
**Rating:** 7
**Confidence:** 4

**Summary:**

Current personalized T2I models, like DreamBooth, are capable of generating personalized images by fine-tuning on a small set of reference images. However, these fine-tuned models often suffer from robustness issues, such as poor compositional capabilities with the pretrained model concepts and other fine-tuned models.

The paper "Direct Consistency Optimization for Robust Customization of Text-to-Image Diffusion Models" introduces a novel fine-tuning method called Direct Consistency Optimization (DCO) to enhance the robustness of such personalized text-to-image (T2I) diffusion models.

DCO introduces a novel training objective that minimizes the deviation between the fine-tuning and pretrained models by optimizing a consistency function. The method involves computing a loss function that measures the deviation in noise prediction errors between the fine-tuned and pretrained models, thus allowing efficient implementation using a noise-prediction loss approach.

**Strengths:**

* Novel Objective Function: The introduction of Direct Consistency Optimization (DCO) represents a novel approach to fine-tuning text-to-image diffusion models. Unlike previous methods that primarily focus on incorporating additional datasets to retain pretrained knowledge, DCO directly controls the deviation from the pretrained model
* The paper effectively combines concepts from constrained policy optimization and noise prediction to derive the DCO loss function. This is a novel contribution based on my knowledge
* good amount of experiments and results visualization
* this work could have big potential impact how the booming application of personalized t2i diffusion models

**Weaknesses:**

* Incremental Improvement: While the DCO method introduces a novel fine-tuning objective, the actual improvement seem marginal (based on figure 4 and figure 7)

* Limited Diversity of Experiments: The experiments primarily focus on subject and style personalization within specific datasets. A broader range of experiments, including diverse set of human images.

* Reproducibility Concerns: Although the methodology is well-explained, the paper does not provide detailed information on implementation specifics such as the codebase, computing resources, or detailed training schedules. This might pose challenges for reproducibility and practical adoption by the community.

**Questions:**

have you thought about trying a kandinsky2.2/dalle2 like model which can directly condition on the image embedding for a (or multiple) reference images?

**Limitations:**

yes its adequately address in line 632

---

> ### Author Rebuttal · Authors · 2024-08-07
>
> Dear reviewer ayXu,
>
> Thank you for valuable comments and suggestions in reviewing our work. We address each of your questions and concerns individually as follows.
>
> ---
> ### [W1] Improvement by using DCO.
> We claim that DCO uses a novel fine-tuning that pushes the frontier of the pareto curve of image similarity and prompt similarity, outperforming previous baseline methods DreamBooth and DreamBooth with prior preservation loss, as well as their variants, e.g., early stopping or different prior preservation loss weights (e.g., see Figure. 7). Not only that, DCO enables direct merging of two independent trained LoRAs, where DreamBooth trained models often fail to preserve the subject or style fidelity during merging.
>
> ---
> ### [W2] Other experiments
> To show the effectiveness of our method, we provided 1-shot subject personalization experiments as shown in Figure.14 and Figure.15 of Appendix. By using a single subject image, our method can generate diverse personalized images that change the style (e.g., photo to 2d animation style, or 3D animation style to photo) or visual attributes (e.g., actions or outfits) by using textual prompts. Specifically, as shown in Figure. 14, we show the capability of our method in learning from a single human image to stylize or change actions. We believe our method can be further explored for human specific datasets, which we leave for future work.
>
> ---
> ### [W3] Reproducibility
> For reproducibility of our method, we provide implementation details in Appendix D, where we describe the compute resources and training schedules. For our codebase, we use PyTorch and Huggingface diffusers library for the implementation. We will release our code in the final manuscript.
>
> ---
> ### [Q1] Personalization of image-conditioned models
> We believe our method can be utilized for personalization of image-conditioned diffusion models (e.g., Kandinsky 2.2 or Dalle-2). Similar to fine-tuning with DCO objective for text-to-image diffusion models, one can consider fine-tuning image conditioned diffusion models that use image embeddings. Specifically, given a pair of reference images (img1 and img2) that share the concept (e.g., subject or style), one can fine-tune the diffusion model on img1 conditioned with image embeddings from img2. Here, DCO loss could be used to restore the pretrained knowledge, which we believe would be effective to prevent overfitting and improve compositional generation of personalization image synthesis. We believe it is an interesting direction, and leave it for future work.

---

### Official Review · Reviewer_aXKh · 2024-07-12

**Soundness:** 2
**Presentation:** 3
**Contribution:** 2
**Rating:** 5
**Confidence:** 3

**Summary:**

This paper introduces a novel fine-tuning objective called Direct Consistency Optimization, which regulates the deviation between fine-tuning and pre-trained models to preserve pre-trained knowledge during the fine-tuning process. Models fine-tuned using this method can be merged seamlessly without interference.

**Strengths:**

1. The results seem good and surpass the baseline methods.
2. The direct merging of subject and style is interesting.

**Weaknesses:**

1. The comparison with Lora is missing. Usually the results of Lora customization are very competitive.
2. In fig1, the pose of the bear in different images are mostly the same as those in the reference image, which is overfitting. It needs to compare the diversity of generated images.

**Questions:**

As seen in weakness.

**Limitations:**

The manuscript includes discussions on limitations and social impacts.

---

> ### Author Rebuttal · Authors · 2024-08-07
>
> Dear reviewer aXKh,
>
> Thank you for valuable comments and suggestions in reviewing our work. We address each of your questions and concerns individually as follows.
>
> ---
> ### [W1] Experiment with LoRA
> We clarify that all our experiments were conducted using LoRA. This has been explicitly mentioned in our manuscript, experimental setup section (e.g., see line 205 and line 242).
>
> ---
> ### [W2] Diversity of generated images.
> Our method is able to generate personalized subject images of various poses, e.g., see Figure. 19 and Figure. 20 in Appendix. Since our method is better than previous methods in terms of prompt fidelity and subject fidelity, one can provide prompts to change the poses or actions to generate diverse images while preserving identities.

---

### Official Review · Reviewer_HfJW · 2024-07-12

**Soundness:** 2
**Presentation:** 2
**Contribution:** 2
**Rating:** 5
**Confidence:** 3

**Summary:**

This paper studies the catastrophically forgetting issue in personalizing text-to-image diffusion model. The main comparable baseline is DreamBooth, which prevents forgetting by finetuning the model on a subsample of original training data while learning new concepts. The proposed method, on the other hand, directly regularizes the deviation of reference model from original model, which resulting a new objective function that essentially penalizing overfitting to the ‘hard examples’ during personalization. Empirical evaluations are conducted on subject and style transfer datasets, where the proposed method achieves superior pareto optimality on image fidelity v.s. prompt following ability, as well as better visual results compared with DreamBooth baseline.

**Strengths:**

1. The proposed method is conceptually simple yet yield strong empirical results (if judged solely based on the displayed results)
2. One benefit of DCO is that it relies solely on the reference data, as the user does not have to gain access to the original training data for regularization during finetuning.
3. The proposed method, in particular DCO, has its novelty and value to the community, if the strong performance can be further verified by extra experiments requested in the weakness section.

**Weaknesses:**

The empirical evaluation can benefit from extra validations.

1. Lacks human evaluation. The paper could benefit from a proper human study from third-party judgers. Currently, the empirical comparisons seem to be confined by automatic evaluation and a subset of visualizations.
2. Compounded factors are not studied separately. The Textual Inversion and comprehensive captioning techniques are deployed on top of DB and DCO. With these enabled, it is unclear what might be the interplay between these techniques and different baselines. For instance, the compatibility of DB might be less compatible with DB than DCO, since DCO only requires the reference data during training. These ablations would not necessarily diminish the proposed method; They are just very beneficial for research purposes IMO.

**Questions:**

1. At a high level, both DCO and DBpp are trying to learn the concept while keeping certain aspects of the new model closer to the original model. It seems (e.g. line 39 and Figure 2) that the motivation for using is based primarily on empirical performance. Could the author elaborate more on the intuition of why DBpp hurts the subject fidelity?

**Limitations:**

Yes

---

> ### Author Rebuttal · Authors · 2024-08-07
>
> Dear reviewer HfJW,
>
> Thank you for valuable comments and suggestions in reviewing our work. We address your questions and concerns as follows.
>
> ---
> ### [W1] Human evaluation
> Following your suggestion, we conduct a user study to compare DCO (ours) against DreamBooth (DB) and DB with prior preservation loss (DB+p.p.) on subject personalization tasks. As in Section 5.1, we train three models (DCO, DB, DB+p.p.) per subject with the identical experimental setup (i.e., all models are trained with comprehensive caption and textual inversion, and compare by generating images with the same random seed. We asked the participants to choose the best one out of three images from each model. Raters are asked to compare images on the following three criteria:
> Subject fidelity: Which image most accurately reproduces the identity (e.g., item type and details) of the reference item?
> Prompt fidelity: Which image most closely aligns with the given prompt?
> Image quality: Which image exhibits the highest quality (e.g., overall visual appeal and clarity)?
> We asked 22 users to evaluate 60 sets of images, resulting in a total 1320 responses per query. The table below shows the result.
>
> Table 1. Subject fidelity, prompt fidelity and image quality user preference.
>
> |                    | Subject fidelity | Prompt fidelity | Image quality |
> |---------------------|----------------------------|---------------------------|-------------------------|
> | DB              | 47.7%               | 22.6%             | 27.7%            |
> | DB+p.p.     | 6.0%                 | 27.6%              | 25.5%            |
> | DCO (ours) | 46.3%               | 49.8%              | 46.9%            |
>
> We see that DCO (ours) largely outperforms DB and DB+p.p. in prompt fidelity and image quality, while showing comparable performance with DB on subject fidelity. This result is consistent with Figure. 1 of our manuscript, where we have shown that DCO outperforms others in image-text similarity (i.e., prompt fidelity), while exhibiting comparable performance in image similarity (i.e., subject fidelity). Interestingly, we find that the users find DCO-generated images are of better quality than others, which demonstrates that DCO mitigates image quality degradation, which often happens when fine-tuning from a few, low quality reference images.   We will include our results as well as the detailed information on our user study format in the final revision.
>
> ---
> ### [W2] Ablation study
> As we mentioned in our manuscript (e.g., line 210-211), using comprehensive caption (instead of compact caption used in DreamBooth paper) improves the performance of both DreamBooth and DCO methods. To support our claim, we provide an additional ablative study on the effect of comprehensive caption. We select 10 subjects from DreamBooth dataset and compare with compact captions on both DreamBooth (DB) and DCO, using the same experimental setup as in Section 5.1. For compact captions, we do not use rare token identifiers and learn textual embeddings as in Section 5.1 Figure A in our attached file shows the Pareto curves with consistency guidance of varying scales ($\omega_{\textrm{con}}$=2.0, 3.0, 4.0, 5.0). We observe that a comprehensive caption (solid line) forms an upper-right frontier compared to compact caption (dashed line) for both DB and DCO.
>
> ---
> ### [Q1] Prior preservation loss hurts subject fidelity
> We hypothesize that learning from both reference and synthetic images as in prior preservation loss hurts subject fidelity due to the concept leakage from the synthetic images. Note that such a behavior (e.g., losing subject fidelity at the cost of generation diversity) has been already observed from the DreamBooth paper (e.g., see Section 4.3 and Table 3 of the DreamBooth paper). On the other hand, DCO preserves the prior without synthetic data by minimizing the shift from the pretrained model while fitting on reference images.

---

> > ### Comment · Reviewer_HfJW · 2024-08-14
> > **Reply to authors**
> >
> > Thank you for the responses, they addressed my concerns. I've improved my ratings accordingly. I suggest the authors to include the new experiments in the revision.

---

> > > ### Author Response · Authors · 2024-08-14
> > >
> > > Thank you for your positive feedback and for updating your ratings. We appreciate your suggestion and will incorporate the details of the additional experiments in our final revision. Thank you once again for your valuable input!

---

### Author Rebuttal · Authors · 2024-08-07

Dear reviewers and AC,

We sincerely appreciate your valuable time and effort spent reviewing our manuscript.

As reviewers highlighted, we believe our work presents a novel training objective that is simple and effective (HfJW, aXKh, ayXu, XWYq) that is supported by qualitative and quantitative experimental results (HfJW, aXKh, ayXu).

We kindly ask you to check the attached supplementary PDF file when acquiring the information provided in the rebuttal comments. Please let us know if you have any comments or concerns that we have not addressed up to your satisfaction.

---

### Decision · Program_Chairs · 2024-09-25

**Decision:**

Accept (poster)

**Comment:**

In this work, authors propose an algorithm called direct consistency optimization where the core idea revolves around an objective that minimizes the deviation between the fine-tuning and pre-trained models by optimizing a consistency function. Overall, the reviewers tend to like the novelty present in this idea and are also encouraged by the potential amount of impact this algorithm can have. There are some concerns over computational cost increase, however, in my opinion, that is not prohibitively expensive for this algorithm to be adapted. Overall, I recommend acceptance.